# Online Rubrics Elicitation from Pairwise Comparisons

**MohammadHossein Rezaei** [1]   **Robert Vacareanu** [1]   **Zihao Wang** [1]   **Clinton Wang** [1]
**Bing Liu** [1]   **Yunzhong He** [1]   **Afra Feyza Akyürek** [1]

## Abstract

Rubrics provide a flexible way to train LLMs on open-ended long-form answers where verifiable rewards are not applicable and human preferences provide coarse signals. Prior work shows that reinforcement learning with rubric-based rewards leads to consistent gains in LLM post-training. Most existing approaches rely on rubrics that remain static over the course of training. Such static rubrics, however, are vulnerable to reward-hacking type behaviors and fail to capture emergent desiderata that arise during training. We introduce Online Rubrics Elicitation (OnlineRubrics), a method that dynamically curates evaluation criteria in an *online* manner through pairwise comparisons of responses from current and reference policies. This online process enables continuous identification and mitigation of errors as training proceeds. Empirically, OnlineRubrics approach yields consistent improvements of up to 8% over training exclusively with human-written rubrics across AlpacaEval, GPQA, ArenaHard as well as the validation sets of expert questions and rubrics. We qualitatively analyze the elicited criteria and identify prominent themes such as transparency, practicality, organization, and reasoning.

## 1. Introduction

Recent advances in reinforcement learning are reshaping the traditional post-training recipe. The work of Guo et al. (2025) demonstrated that supervised fine-tuning on instructions can be skipped altogether, with policies (e.g. R1-Zero) trained directly via reinforcement learning, disrupting the way researchers think about post-training. Since then, much of the focus has shifted towards reinforcement learning.

However, R1-Zero was trained only using verifiable rewards; the final response is easily gradable, think of a number or code snippet with unit tests, which is only applicable to limited domains.

To accommodate broader settings, rubric-based scoring for reinforcement learning emerges as an alternative way for reward modeling, particularly for long-form responses (Viswanathan et al., 2025; Gunjal et al., 2025; Huang et al., 2025; Anugraha et al., 2025). Rubrics are comprised of a list of input-specific criteria that characterizes an ideal response; one example criterion in the finance domain is *"States shocking basis causes nonlinear effects in margin calls"*. Each criterion has an importance weight: satisfying positively weighted criteria yields reward, while satisfying negatively weighted criteria yields penalty. During training, an LLM-based grader evaluates a response against each criterion in the rubric, producing binary satisfaction scores; and the overall score is the weighted average of these grades. This framework extends reinforcement learning to both verifiable and non-verifiable aspects of responses, spanning generalist and expert domains.

Rubrics often emphasize the desired behaviors with less coverage of undesired properties. *Offline rubrics* created a priori, human-written or synthetic, cannot realistically cover every unexpected (and desired) pattern. Fixed checklists (Wang et al., 2024) to enforce generally helpful patterns e.g. truthfulness, instruction following or relevance, fall short in preventing nuanced errors. For example, Huang et al. (2025) identifies "self-praising" as one emerging pattern during reinforcement learning from rubrics, think of including *"The following advice is the most relevant"* as part of the response; these praises often fool the LLM-based verifier into believing that the given advice is indeed relevant. Such patterns are especially difficult for generic "catch-all" rubrics to capture when they are sample-specific. Moreover, correct traits in some generations can go unnoticed if not readily rewarded by the existing rubrics.

We introduce OnlineRubrics, a framework for eliciting evaluation criteria dynamically via pairwise comparisons. OnlineRubrics leverages a pair of responses in creating additional criteria where the responses are sampled from the current policy and a control model. Our work, as depicted

---

[1]Scale AI, San Francisco, United States. Correspondence to: MohammadHossein Rezaei <mohammad.rezaei@scale.com>.

*Proceedings of the 43rd International Conference on Machine Learning*, Seoul, South Korea. PMLR 306, 2026. Copyright 2026 by the author(s).

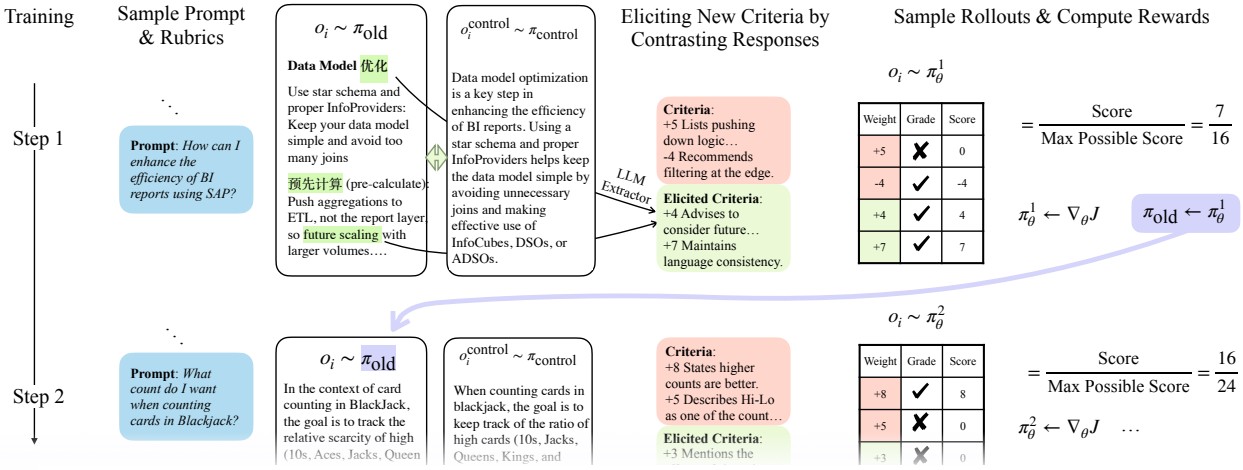

*Figure 1.* At any step during training, OnlineRubrics starts off by considering a pair of responses, one of which is from the current policy before updates and another from a *control* model e.g. reference model. We follow with LLM-based rubrics elicitation and deduplication steps to generate a set of criteria. These criteria along with existing criteria (e.g. human-written or synthetic) are used to create the reward in the policy gradient algorithm.

in Fig. 1, is inspired by the large body of literature on preference learning (Akrour et al., 2011; Fürnkranz et al., 2012; Schoenauer et al., 2014) and pairwise reward modeling (Christiano et al., 2017; Stiennon et al., 2020; Ouyang et al., 2022). While LLMs are imperfect judges of quality (Gu et al., 2024), we found that pairwise comparisons are easier to make for the models when identifying new criteria than directly making a quality assessment or creating new criteria by considering a single response (point-wise elicitation). The additional criteria simply augments the existing rubric, enabling seamless integration of OnlineRubrics with any rubric-based scoring mechanism.

In training and evaluating our approach, we curate two datasets for expert (scientific use-cases) and generalist domains. We additionally conduct out-of-distribution evaluations using public benchmarks, comparing different approaches to reward estimation. OnlineRubrics results in absolute gains of up to 25% over the initial instruct model across various benchmarks including GPQA-Diamond, GSM8K, AlpacaEval, and Arena-Hard.

Our main contributions are: (i) we introduce OnlineRubrics, an online rubric elicitation method that derives new criteria directly from pairwise contrasts between the current policy and a control policy during RL training, and we contrast it against pointwise elicitation; (ii) we curate two rubric datasets (Generalist and Expert) spanning generalist and expert (Physics, Chemistry, Biology, Math) domains; and (iii) we provide a comprehensive empirical study showing consistent improvements across base model families (Qwen and Llama) and model sizes (3B–14B) on multiple benchmarks

and a blind human validation study.

## 2. Related Work

**Reward Modeling** The dominant paradigm in LLM alignment is to learn a reward function from feedback. Foundational work in Reinforcement Learning from Human Feedback (RLHF) established the use of pairwise preference comparisons–preferred over less robust pointwise scores–to train an explicit reward model (Ouyang et al., 2022; Stiennon et al., 2020). This process was later simplified by methods like Direct Preference Optimization (DPO; Rafailov et al. (2023)), which optimizes policies directly on preference data. Methods for generating feedback have also advanced: Bai et al. (2022) pioneered the use of AI feedback (RLAIF) by leveraging a fixed set of principles for model self-feedback. More recently, research has focused on improving the reward model's intrinsic capability. (Liu et al., 2025b) established inference-time scaling laws for generalist reward models, while (Whitehouse et al., 2025) incentivizes faithful evaluation by training *thinking* LLM judges. (Lu et al., 2025) proposes adaptively adjusting reward weights over a set of competing objectives such as accuracy and efficiency during online learning. These works collectively highlight the need to optimize towards a collection of desired outcomes and to do so adaptively and in an online manner.

**Verifiable Rewards** While preference-based rewards provide flexible but often fuzzy signals, verifiable rewards offer exact supervision whenever the outcome can be automat-

ically checked. Reinforcement Learning with Verifiable Rewards (RLVR) improves reasoning by optimizing policies against automatically checkable outcomes, such as numeric answers or unit-tested code. Recent work has shown its effectiveness across various domains: DeepSeek-R1 (Guo et al., 2025) and General-Reasoner (Ma et al., 2025) achieved strong results on benchmarks such as GSM8K (Cobbe et al., 2021), MMLU (Hendrycks et al., 2021), and GPQA (Rein et al., 2024). In medicine, Zhang et al. (2025) enabled a 3B model to reach expert-level performance. Foundational studies confirm that RLVR incentivizes correct reasoning processes, not just correct answers (Wen et al., 2025). Despite these strengths, RLVR does not readily extend to long-form answers.

**Evaluating and Training with Rubrics** Recent work has extended the concept of verifiable rewards from domains like math and coding to more open-ended tasks by using rubrics for structured evaluation. This rubric-based approach has been adopted in various benchmarks for both expert (Arora et al., 2025; Starace et al., 2025) and generalist domains (Deshpande et al., 2025). Beyond evaluation, rubrics are now increasingly used as direct reward signals. Using structured rubrics as a direct reward has proven effective in both expert reasoning (Gunjal et al., 2025) and generalist alignment (Viswanathan et al., 2025). A diverse set of rubrics has also been used to train a single, robust reward model that generalizes across various domains (Anugraha et al., 2025). Closer to our setting, Liu et al. (2025a) generate synthetic rubrics from instructions via contrastive comparison of preferred and rejected responses and use them to train rubric-conditioned reward models in an offline setting; unlike them, we elicit rubrics in an *online* manner directly from pairwise comparisons of current and control policy rollouts during RL training. Wei et al. (2026) mine question-specific rubrics from online sources and use them as reward signals, but the rubrics remain fixed throughout training. Concurrently, Dr Tulu (Shao et al., 2025) generates rubrics in an online manner from model rollouts in a free-form way while having access to external knowledge for deep research; OnlineRubrics, in contrast, elicits new criteria from pairwise comparisons against a control policy rather than from an external knowledge source, yielding more grounded rubrics that focus on emerging behaviors during training. Our work complements these methods; instead of using a static rubric or training a rubric-agnostic model, OnlineRubrics dynamically augments criteria to adapt to the policy's emergent behaviors.

# 3. Background

Rubric-based scores are often used as drop-in replacement for rewards in any policy gradient learning algorithm.

## 3.1. Training Setup

In this work, we used GRPO algorithm (Shao et al., 2024) which maximizes the following objective

$$\mathcal{L}_{\text{GRPO}}(\theta) = \mathbb{E}_{i \sim \mathcal{D}, j \sim \mathcal{G}_i} \Big[ \min \big( r_{i,j}(\theta) \, \hat{A}_{i,j}^{\text{group}}, $$
$$\text{clip}(r_{i,j}(\theta), 1-\epsilon, 1+\epsilon) \, \hat{A}_{i,j}^{\text{group}} \big) - \beta \mathbb{D}_{\text{KL}}(\pi_\theta || \pi_{\text{ref}}) \Big] \quad (1)$$

where $r_{i,j}(\theta) = \frac{\pi_\theta(o_{i,j}|x_i)}{\pi_{\theta_{\text{old}}}(o_{i,j}|x_i)}$ is the probability ratio, and advantages are calculated as normalized rewards:

$$\hat{A}_{i,j}^{\text{group}} = \frac{R_j - \text{mean}(\mathbf{R})}{\text{std}(\mathbf{R})} \, . \quad (2)$$

$\mathcal{D} = \{x_i, \mathcal{C}_i\}$ is the set of training prompts and criteria, $j$ indexes the output samples $o_j$ from the group $o_j \sim \mathcal{G}_i$, $\pi_{\theta_{\text{old}}}$ is the policy before the update, $\pi_\theta$ the target policy. The rewards are computed independently for each $o_j$ in the group and denoted by $\mathbf{R} = \{R_1, R_2, \ldots, R_G\}$ where $G$ is the group size. In this work, we will assume that the true reward $U$ can be modeled as a function of some latent criteria and argue in Section 4.2 that for optimal modeling of the true reward all criteria should be elicited.

## 3.2. Rubric Based Rewards

In RLHF, reward signals in LLM training are traditionally modeled after human preferences with an explicit reward model in PPO (Schulman et al., 2017) and GRPO or implicitly in DPO. In the case of queries where quick verification of the final answer is possible (i.e. numeric or short answer), exact match replaces human preferences for reward. More recently, rubrics for evaluating long-form answers are being used for calculating final scores (Gunjal et al., 2025; Huang et al., 2025; Viswanathan et al., 2025) where an LLM-based grader (denoted by $\text{LLM}_{\text{grader}}$) evaluates a response against each criteria to compute $R_j$ in Equation (3):

$$R_j = q\Big(\text{LLM}_{\text{grader}}\big(o_j, x_i, \mathcal{C}_i\big)\Big) \quad (3)$$

where $\mathcal{C}_i = \{(c_1, w_1), (c_2, w_2), \ldots, (c_d, w_d)\}$ is a collection of criteria with corresponding importance weights that describe an ideal response to the prompt, and $q$ is an aggregation function. The judge $\text{LLM}_{\text{grader}}$ (Zheng et al., 2023) evaluates the output $o_j$ against each criterion in $C_i$ and produces a list of binary outcomes which are then reduced to a single scalar value by $q$ using the weights, if applicable. We implement the reduction function as a weighted sum of the grades normalized by the total possible score:

$$q(x, o, \mathcal{C}) = \frac{w^\top \text{LLM}_{\text{grader}}(x, o, \mathcal{C})}{\sum_{k:w_k > 0} w_k} \quad (4)$$

```
You are given a prompt and pair of responses to the same prompt. Your task is to identify their
differences not already covered by the existing rubrics. [truncated] First, analyze both responses
to identify the differences. Then, transform these observations into new evaluation criteria if
they're not already covered by existing rubrics. This is very important: any rubric that you
introduce should be based on one of the responses. Do not use your own knowledge to introduce new
criteria that are not based on one of the responses.

Focus on criteria that distinguish genuinely helpful responses from those gaming the system.
[truncated] Assign a positive weight (integer) to each of the new criteria based on the relative
importance of the criterion to the existing criteria.
If no meaningful new criteria are needed, return an empty list.

{{Existing Rubric}}
{{Response A}}
{{Response B}}
```

*Figure 2.* Abbreviated system prompt template used for LLM$_{\text{extractor}}$, see full prompt in Fig. 8.

---

**Algorithm 1** Online Rubric Eliciting (OnlineRubrics)

---

**Require:** Policy $\pi_\theta$, control policy $\pi_{\text{control}}$, dataset $\mathcal{D}$, extraction prompt $P_e$, hyperparameter $M$

  **for** $step = 1, 2, \ldots, N$ **do**

    Sample prompts and criteria $\{x_i, \mathcal{C}_i\}$ from $\mathcal{D}$

    Update $\pi_{\text{old}} \leftarrow \pi_\theta$

    Generate $M$ candidate responses $\{o_{i,j}\}$ using $\pi_{\text{old}}$

    Generate $M$ candidate responses $\{o_{i,j}^{\text{control}}\}$ using $\pi_{\text{control}}$

    Initialize $C_i^e \leftarrow \emptyset$

    **for** $k = 1, 2, \ldots, M$ **do**

      Extract new criteria and augment:

      $C_{i,k}^e \sim \text{LLM}_{\text{extract}}(x_i, o_{i,k}, o_{i,k}^{\text{control}}; P_e)$

      $C_i^e \leftarrow C_i^e \cup C_{i,k}^e$

    **end for**

    De-duplicate $C_i^e$

    Compute rewards using Equation (3) and $\mathcal{C} = \mathcal{C}_i \bigcup C_i^e$

    Compute group advantages $\hat{A}_{i,j}$ Equation (2)

    Update $\theta$ via policy gradient by maximizing Eq. 1

  **end for**

---

where $\text{LLM}_{\text{grader}}\ (x, o, \mathcal{C}) \in \{0,1\}^d$ is the binary grades corresponding to each criterion.

# 4. Online Rubric Elicitation

Rubric-based reward calculation provides richer feedback than reward-model-based post-training, yet it fails to mitigate the problems that might emerge during policy gradient updates. Specifically, we observe that initial rubrics tend to represent the desired qualities of an ideal response while putting less emphasis on describing undesired qualities. For example, when the prompt is *How can I travel to San Francisco from San Jose?* and the rubric is *(+9, The response recommends Caltrain)* both responses *Consider Caltrain or rent-a-car, however, Caltrain doesn't serve East Bay* and *Caltrain and renting a car are both good options* get the full score while the former has redundancy about Caltrain's

East Bay coverage. Such mishaps may only be detected as they arise during rollouts. Moreover, emerging qualities that are not currently rewarded by the existing rubric set will be overlooked by the algorithm.

We propose a novel method called OnlineRubrics that leverages pairwise comparison of candidate responses to derive novel criteria–OnlineRubrics is designed to capture potential errors and identify useful features. Having derived insights from the pairwise reward modeling literature (Bradley & Terry, 1952; Stiennon et al., 2020; Ouyang et al., 2022), this approach simply augments the set of offline criteria i.e. the portion of the rubric that is created a priori for the specific prompt, with more criteria derived during the training. Our approach is different from recent work that uses a fixed set of criteria (Anugraha et al., 2025) for multiple data points or other procedures to extract rubrics in a pointwise manner by simply considering a prompt (Huang et al., 2025).

## 4.1. LLM-based Criteria Elicitation

OnlineRubrics begins with an initial set of *offline criteria* $\mathcal{C}_i$ that may be provided by human annotators or created synthetically. During policy training, at step $t$ before any updates, given a prompt $x_i$ we sample a set of candidate responses from a *control* policy (e.g. the initial policy, $\pi_{\text{ref}}$, or the policy from the previous step $\pi_{\text{old}}$) and the current policy $\pi_\theta^t$. We define an LLM-based rubric extractor LLM$_{\text{extractor}}$ conditioned on the system prompt $P_e$ (see Figure 2) whose task is to identify the differences between a pair of responses $(o_{i,j}, o_{i,j}^{\text{control}})$ sampled from the current and control policies, respectively, and turn them into useful criteria and corresponding weights. LLM$_{\text{extractor}}$ is instructed to provide references within the responses to where the difference appears. We repeat this procedure independently for each prompt in the batch and augment their corresponding rubrics with the new criteria before the policy parameter update. We provide the procedure in Algorithm 1.

We adopt a two-step approach for criteria elicitation; in the

first step, we ask $\text{LLM}_{\text{extractor}}$ to enumerate the meaningful differences between a pair of responses with references to where these differences arise in the responses. In the second stage, we reduce the criteria that are duplicates or overlap significantly to avoid redundancy following our desiderata Section 5. The system prompt template used to extract rubrics is given in Figure 2 and the deduplication prompt is available in Figure 9. By default, we compare eight pairs of rollouts from each of the control and current policies and extract about eight criteria at the end of the procedure. Note that the elicited criteria are not persistent across steps and are used during the current step only. LLM-based criteria elicitation adds a computational overhead to the training process. However, this overhead is comparable to or less than, verifying responses using LLM-based graders. For a detailed computational cost analysis, see Appendix A.

**OnlineRubrics Variants**   We experiment with two variants for where we change the source of alternative responses $\pi_{\text{control}}$ with one of $\pi_{\text{ref}}$ or $\pi_{\text{old}}$. We hypothesize that setting sampling the responses from the $\pi_{\text{old}}$ makes the reward-hacking behaviors less likely to surface, since both candidate and control rollouts are sampled from the same distribution, hence subject to the same failure modes. That said, we empirically observe in Table 2 that this version also performs quite strongly compared to the setting $\pi_{\text{control}} = \pi_{\text{ref}}$.

### 4.2. A Formal Motivation for OnlineRubrics

Let $f$ be the grades from $\text{LLM}_{\text{grader}}$ for the prompt, response and criteria triplet $(x, o, \mathcal{C})$ such that $f(x, o, \mathcal{C}) \in \{0, 1\}^d$ where $\mathcal{C}$ and $w$ are the set of criteria and weights and $d$ is the size of the criteria. Let also $\mathcal{C}^E$ (explicit) and $\mathcal{C}^I$ (implicit) to denote to the set of criteria in the rubric and those not in the rubric, respectively, and $f_E(x, o)$ to indicate the binary grades for the output $o$ under criteria $\mathcal{C}^E$.

*Proposition* 1. Suppose that

- $\mathcal{C}^*$ is the set of true criteria. $f_*$ can be split into $f_* = (f_E, f_I)$ and $\mathcal{C}^* = (\mathcal{C}^E, \mathcal{C}^I)$.
- The true reward is $U(x, o) = w_E^\top f_E(x, o, \mathcal{C}^E) + w_I^\top f_I(x, o, \mathcal{C}^I)$ and the estimated reward $R_t(x, o) = w_E^\top f_E(x, o)$ at step $t$.
- Assuming GRPO style updates, the gradient under the true reward then would be $g_U = \mathbb{E}[\nabla_\theta \log \pi_\theta(o|x) U(x, o)]$ and the estimated gradient $g_{R_t} = \mathbb{E}[\nabla_\theta \log \pi_\theta(o|x) R_t(x, o)]$

Then,

$$\|g_U - g_{R_t}\|_2 \leq \sqrt{\mathbb{E}\left[\left\|\nabla_\theta \log_{\pi_\theta}\right\|^2\right]} \|w_I\|_1$$

Proposition 1 shows that the difference between the gradient steps is upper-bounded by $\|w_I\|_1$ times the expected squared

*Table 1.* Generalist and Expert Rubrics datasets statistics.

| | Train | | Eval. | |
|---|---|---|---|---|
| | # Sam. | # Rub. | # Sam. | # Rub. |
| Generalist | 1,500 | 15,528 | 487 | 5,003 |
| Expert | 1,864 | 33,554 | 332 | 5,938 |
| Math | 584 | 9,512 | 104 | 1,688 |
| Biology | 506 | 9,863 | 90 | 1,750 |
| Physics | 314 | 5,631 | 56 | 1,001 |
| Chemistry | 460 | 8,548 | 82 | 1,499 |

norm of the policy score function. Augmenting the rubric to better approximate the true criterion set leads to better estimation of the true gradient hence improved stability and sample efficiency during training. That said, OnlineRubrics should be viewed as a step toward tightening the upper bound on the implicit, unmodeled mass $\|w_I\|_1$, rather than a complete recovery of the true criteria set.

While this analysis is not intended to provide asymptotically tight guarantees for rubric-based policy learning, it motivates why eliciting missing criteria matters in rubric-based reward modeling. We empirically show in Section 6.3 that a fixed of rubrics describing general qualities of a good response such as correctness or conciseness (*Universal Requirements*) is not sufficient to describe the above-mentioned unmodeled mass. Proof for Proposition 1 is given in Appendix B.

## 5. Datasets

We trained OnlineRubrics with two collected rubric datasets: Generalist Rubrics and Expert Rubrics. Generalist Rubrics consists of real-world, single-turn prompts contributed with user consent and curated to be safe, rubric-eligible, and generalist in scope. For each prompt, human annotators authored a prompt-specific rubric composed of weighted, binary-checkable criteria.

Expert Rubrics extends the same rubric framework to expert-authored problem sets across Physics, Chemistry, Biology, and Math. Each task bundles a prompt, an expert grading rubric with binary-evaluable and weighted criteria, sample model responses, and detailed rubric ratings. We use a subset of both datasets as evaluation sets and exclude from training. Table 1 shows the statistics of the datasets. On average, Generalist set contains 10.4 rubrics per sample and Expert set contains 18.0 rubrics per sample. Across both datasets, rubrics are human-written and follow the same annotation principles: criteria should be *Atomic*, *Descriptive* (i.e. all information necessary for grading is available) and *Objective*; ensuring they can be verified reliably and used as dense reward signals in offline and online training. See Appendix C for data samples.

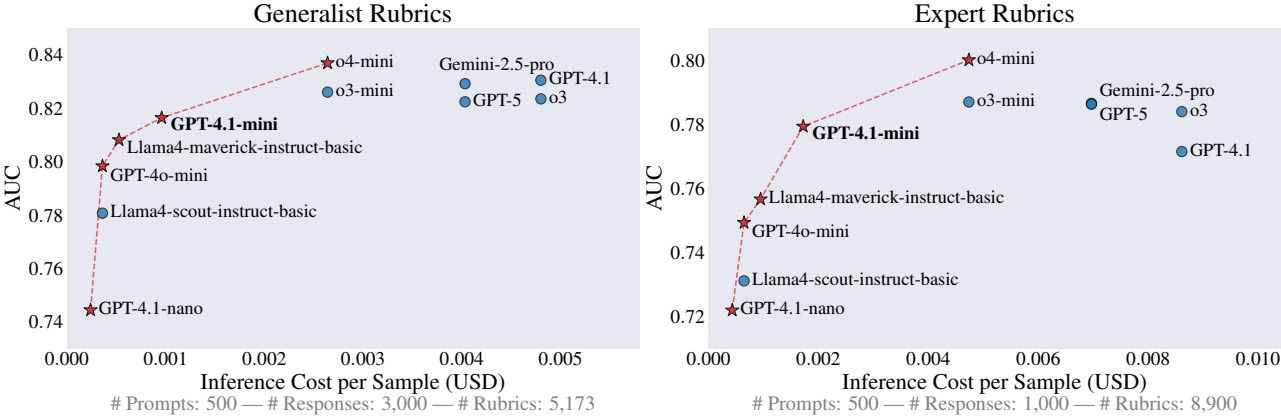

*Figure 3.* Performance of different LLM graders. AUC score is calculated using the receiver operating characteristic (ROC) curve. The best grader is the one with the highest AUC score and the lowest inference cost per sample. Models on the Pareto frontier (red dotted line) are the best trade-off between the two metrics. We use GPT-4.1-mini as our grader, balancing alignment quality with inference cost.

We evaluate OnlineRubrics on (1) evaluation sets of both datasets by calculating rubrics score and win rate using Gemini 2.5 Pro (Comanici et al., 2025) as an LLM-Judge, and (2) on the following public benchmarks: GPQA-Diamond (Rein et al., 2024), GSM8K (Cobbe et al., 2021), AlpacaEval (Li et al., 2023; Dubois et al., 2024)), and Arena-Hard (Li et al., 2024a;b).

# 6. Experiments and Results

We begin by identifying the most effective LLM-based grader for rubric grading in Section 6.1. Next, we introduce our baselines in Section 6.2 and report the main results with OnlineRubrics in Section 6.3. Finally, we perform a qualitative analysis of the elicited rubrics in Section 6.5.

We train Qwen-2.5-7B-Instruct (Qwen et al., 2025) with GRPO as the training algorithm on the train set from both Generalist and Expert Rubrics datasets and evaluate on the validation set of the respective datasets 10 times over each epoch. We use *o3-mini* as the LLM$_{extractor}$ and set the number of pairwise comparisons to 8. Further details on the experimental settings are given in Appendix D.

## 6.1. Grader Selection

Rubrics-based training and evaluation require an LLM grader to evaluate whether output $o_j$ meets criteria $\mathcal{C}_i$ for prompt $x_i$, producing a sequence of binary scores. Because LLMs grade responses differently and human-annotated rubric data is scarce, we collected human grading of the human-written rubrics for 2–6 sampled responses per prompt over 500 prompts in each of the Expert and Generalist sets.

Using this dataset, we evaluate several LLM graders (Figure 3). Because rubrics-based training requires evaluating many rollouts per prompt, a low inference cost per sample

is important. All verifiers perform better on the Generalist set than the Expert set (average AUC 0.811 vs. 0.768); notably, the Pareto frontier is identical across the two datasets, suggesting domain does not affect the relative ranking of verifiers.

## 6.2. Baselines

Previous work establishes rubrics as a strong alternative to preference-based learning such as RLHF (Viswanathan et al., 2025). Building on top of this line of research, we compare our method with the following baselines (with further details in Appendix G):

**LLM-Judge Score** We train the model by using an LLM-judge (default is o3-mini) to grade the responses on a Likert scale without any rubrics. The input to the LLM is a prompt-response pair $(x_i, o_j)$, and the output is a Likert score that is converted to a reward $R_{i,j}$ using a linear mapping.

**Offline Rubrics (Synthetic)** We use the same prompts available in the Generalist and Expert Rubrics datasets. However, instead of using human-written rubrics, we synthetically create rubrics using o3-mini.

**Offline Rubrics (Human)** We train the model with human-written rubrics from the Generalist and Professional Rubrics datasets. As we shall see, this is better than synthetic rubrics.

**Universal Requirements** As discussed in Section 2, previous work has argued that adding a fixed set of criteria to all samples helps in making training more stable and prevent reward hacking. We use the same universal requirements as in Viswanathan et al. (2025) and show OnlineRubrics, which elicits sample-grounded rubrics online, outperforms these universal requirements as a static approach.

**Pointwise Elicitation** In order to show the effectiveness of pairwise comparison, we also extract rubrics point-wise

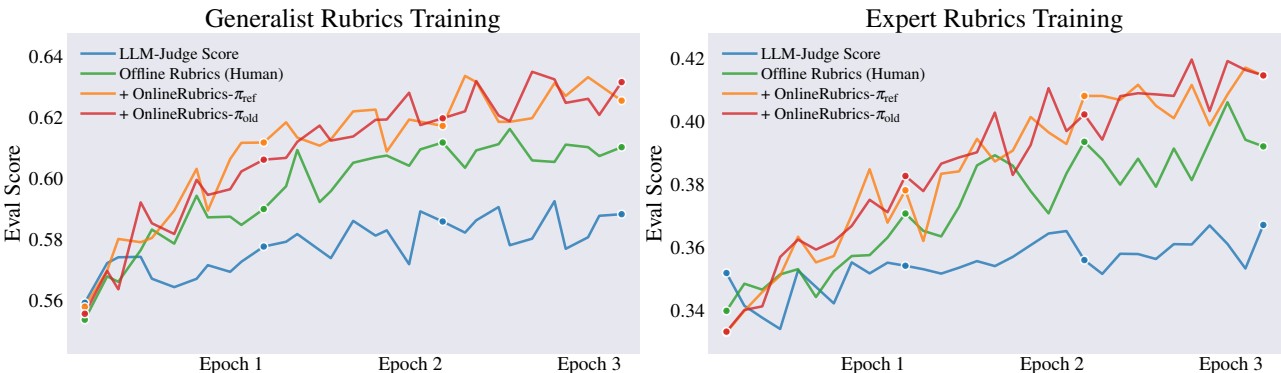

*Figure 4.* Results on the evaluation set of the Generalist and Expert datasets during training. The evaluation set is fixed and contain no elicited rubrics. Both OnlineRubrics methods outperform using Offline Rubrics (Human) or LLM-judge Score (a Likert scale).

using the same extractor model. The input to the extractor is prompt $x_i$, a response $o_j$ from the reference policy, and existing rubrics $\mathcal{C}_i$. The output is a set of criteria $C_i^e$ that we add to the human-written rubrics $\mathcal{C}_i$.

### 6.3. Results and Discussion

Figure 4 shows the training curves for the Generalist and Expert datasets. Training with rubrics consistently scores higher and is more sample efficient than using LLM-Judge scores. More interestingly, adding the elicited rubrics during training (OnlineRubrics) improves the performance of the model on the evaluation sets of both datasets, which contain no elicited rubrics.

Tables 2 and 3 present results on instruction-following and reasoning benchmarks. Offline Rubrics (Human) consistently beats LLM-Judge scores and surpasses synthetic rubrics on 7 of 9 metrics. Adding elicited rubrics on top (OnlineRubrics) further boosts performance across all benchmarks: on AlpacaEval, OnlineRubrics-$\pi_{\text{ref}}$ raises the win rate from 46.4% to 55.0% and LC-WR from 28.0% to 31.5%. We note that the gains in raw win rate are larger than in the length-controlled variant; this is consistent with OnlineRubrics producing somewhat longer responses on average (see App. H), but the LC-WR improvements remain positive across both OnlineRubrics variants, suggesting that the gains are not solely attributable to length.

OnlineRubrics consistently outperforms Universal Requirements, indicating that sample-grounded elicited rubrics are more effective than fixed criteria that fail to capture per-prompt nuances. Pointwise-extracted rubrics often improve over offline rubrics but are still surpassed by OnlineRubrics (48.1 vs. 54.0/55.0 on AlpacaEval, 51.1 vs. 55.7/56.5 on Arena-Hard): pairwise differences highlight discriminative properties that a single response cannot.

**Significance and robustness.** Across three independent training seeds, OnlineRubrics significantly outperforms both

Offline Rubrics and Pointwise Extraction on rubric scores, while the gap between Pointwise Extraction and Offline Rubrics is not significant. OnlineRubrics wins on every single seed. Full per-seed numbers and $t$-test details are in Appendix E.

**Human validation.** To verify that gains are not specific to the LLM judge, we conducted a blind human validation study comparing OnlineRubrics-$\pi_{\text{ref}}$ against Offline Rubrics (Human) on 100 randomly sampled prompts. A human annotator preferred OnlineRubrics in 70 out of 100 cases. Qualitative analysis indicates OnlineRubrics responses were consistently better in depth, contextual precision, use of examples, and completeness; the main failure mode was over-elaboration, where extra length was unnecessary. Details are available in Appendix F.

### 6.4. Ablations

We provide four sets of ablations. First, we experiment with reducing the total number of comparisons per prompt, using 1, 2, 4 and 8 pairs of responses. Second, we consider alternative rubric extractors including an open-source model. Third, we vary the policy size within Qwen2.5 (3B/7B/14B). Finally, we experiment with Llama 3.2 3B and Llama 3.1 8B as base models for generalist and expert rubrics, respectively. Results are provided in Appendix H.

We find in Table 5 that increasing the number of response pairs to elicit new criteria generally improves performance — on Alpaca-Eval, 8 pairs yields 7–10% relative improvement over a single pair — although the final criterion count does not strictly increase since extraction and deduplication are non-deterministic.

Moreover, we find in Table 6 that while the choice of the extractor model does affect performance gains obtained with OnlineRubrics, even lightweight models such as GPT 4.1- Nano are competitive with o3-mini. Further, in certain settings GPT OSS 120B outperforms the Offline Rubrics

*Table 2.* Results on the instruction-following benchmarks. WR stands for Win Rate and LC-WR is Length-Controlled Win Rate. We highlight the best performing model in each column in bold and underscore the second best performing approach. Both OnlineRubrics methods (OnlineRubrics-$\pi_{\text{ref}}$ and OnlineRubrics-$\pi_{\text{old}}$) are consistently better than the baselines except for one case.

| Model | Generalist Rub. | | Alpaca-Eval | | Arena-Hard |
|---|---|---|---|---|---|
| | Score | WR | WR | LC-WR | WR |
| *Baselines* | | | | | |
| Qwen-2.5-7B-Instruct | 55.4 | 39.0 | 30.0 | 28.2 | 50.0 |
| + LLM-Judge Score | 58.8 | 51.3 | 42.2 | 26.9 | 51.0 |
| + Offline Rubrics (Synthetic) | 58.8 | 52.8 | 39.5 | 28.2 | 51.5 |
| + Offline Rubrics (Human-written) | 61.0 | 62.2 | 46.4 | 28.0 | 52.4 |
| + Universal Requirements | 59.4 | 59.1 | 44.4 | 30.3 | 53.8 |
| + Pointwise Extraction | 62.9 | 64.9 | 48.1 | 29.4 | 51.1 |
| *Our Methods* | | | | | |
| + OnlineRubrics-$\pi_{\text{ref}}$ | 62.7 | 67.6 | 54.0 | **31.5** | 55.7 |
| + OnlineRubrics-$\pi_{\text{old}}$ | **63.2** | **68.2** | **55.0** | 30.4 | **56.5** |

*Table 3.* Results for the expert domain. WR stands for win rate and Acc. stands for accuracy. We highlight the best performing model in each column in bold and underscore the second best performing approach. Both OnlineRubrics methods outperform the baselines.

| Model | Expert Rub. | | GPQA-D | GSM8K |
|---|---|---|---|---|
| | Score | WR | Acc. | Acc. |
| *Baselines* | | | | |
| Qwen-2.5-7B-Instruct | 33.6 | 31.9 | 34.7 | 79.2 |
| + LLM-Judge Score | 36.7 | 44.0 | 34.5 | 79.1 |
| + Offline Rubrics (Synthetic) | 37.1 | 46.4 | 36.6 | 79.2 |
| + Offline Rubrics (Human-written) | 39.2 | 51.8 | 36.2 | 79.9 |
| + Universal Requirements | 39.7 | 53.3 | 36.6 | 80.1 |
| + Pointwise Extraction | 40.9 | 57.1 | 33.6 | 78.3 |
| *Our Methods* | | | | |
| + OnlineRubrics-$\pi_{\text{ref}}$ | 41.4 | **61.0** | 37.6 | 80.0 |
| + OnlineRubrics-$\pi_{\text{old}}$ | **41.5** | 56.5 | **38.1** | **80.5** |

(Human) baseline when evaluated in the length-controlled setting in Alpaca-Eval. Finally, Table 8 we replace our base model with two Llama models–while baseline performances compared to the default Qwen model drops significantly, OnlineRubrics improve over the baselines by between 2-5% in generalist and expert domains.

Within the Qwen2.5-Instruct family at three sizes (3B, 7B, 14B), rubric-satisfaction gains over Offline Rubrics (Human) grow with model size ($+1.3\,\text{pp} \rightarrow +2.6\,\text{pp} \rightarrow +3.2\,\text{pp}$), with consistent $+4.6$–$+6.1\,\text{pp}$ Arena-Hard gains (Table 7). This suggests larger policy models within the same family better leverage the elicited criteria.

### 6.5. Qualitative Analysis

We qualitatively analyzed the elicited criteria, contrasting them with the human-written rubrics. We applied an LLM-based comparison of rubric updates (initial vs. last epoch) followed by clustering and manual inspection. Several consistent improvement types emerge: elicited criteria fre-

quently introduce *evidence grounding*, *reproducibility*, and *holistic anti-gaming criteria*, broadening evaluation beyond surface-level correctness; many emphasize *practicality and real-world feasibility*; and meta-criteria such as *structural organization*, *causal reasoning*, and *uncertainty handling* enhance methodological coverage. Overall, online elicitation expands and strengthens rubrics, adapting dynamically as errors emerge. A full cluster list with proportions is in Appendix I.

We also identified extracted criteria that directly target potential reward-hacking. For the prompt *"Give me advice on how to pass the theory exam for minibuses (D1) in Northern Ireland"*, the criterion *"The response should provide region-specific guidance by referencing official materials (e.g., DVTA for Northern Ireland)"* was elicited against a generic-advice response (*misgeneralization of context*). For an occupational-safety prompt, an elicited criterion to *"focus on specified key topics. . . by avoiding extraneous details"* directly mitigates verbosity-as-comprehensiveness hacks.

# 7. Conclusion

We have described OnlineRubrics, a framework for dynamically eliciting new criteria from pairwise comparisons of responses during reinforcement learning. Unlike static rubrics which may be incomplete or become obsolete as training progresses, our approach aims to continuously surface overlooked errors or emerging desired properties. This yields robust gains across expert and generalist domains. Our results show improvements of up to 8 percentage points over training exclusively with human-written rubrics on AlpacaEval, GPQA and Arena-Hard. By moving rubric elicitation online, OnlineRubrics adapts as training evolves, capturing emergent behaviors and strengthening alignment beyond what fixed rubrics allow.

## Limitations

Our proposed technique is reliant on LLM generations which are prone to errors and biases which may propagate into the generated criteria. Similarly, although common practice, our evaluations on the curated eval set, AlpacaEval and Arena-Hard are also LLM-based and are subject to LLM errors. Moreover, online criteria elicitation is an especially costly step as it requires extra rollouts, steps for comparing and de-duplication. The pipeline also relies on proprietary closed-source models at multiple stages (extractor, grader, evaluation judge); while our extractor ablation in Table 6 shows that lightweight open-source models such as GPT OSS-120B are competitive, fully open-source replacements for the grader and judge remain an important direction for reproducibility. By construction, OnlineRubrics elicits criteria grounded in one of the two compared responses; in expert domains where neither response surfaces the correct reasoning, the elicitation procedure has no basis to introduce it, so the method's ceiling is bounded by the quality of the rollouts themselves. To partially mitigate distillation concerns, the extractor is explicitly instructed not to use its own knowledge to introduce new criteria (Figure 8), so any residual knowledge transfer is bounded by the differences already present in the rollouts. Finally, when the most recent policy is used as the control model, rubrics may undergo semantic drift; while we hypothesize that using the reference policy as the control can reduce this drift, future work can directly measure this effect.

## Impact Statement

The proposed procedure involves model training with synthetic AI generated criteria. While the main proposition of the paper is not to claim all such criteria are truthful and some may contain errors, as with any synthetic training data approach this method entails the risk of reinforcing hallucinations.

Moreover, we have worked with human experts in creating our prompts and rubrics. All annotators are independent contractors and were compensated at rates consistent with fair labor practices and applicable local laws. Participation to this study was entirely voluntary, with the option to decline tasks at any time. Datasets do not include any personally identifiable information or sensitive data.

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

## A. Computational Cost Analysis

The computational cost of OnlineRubrics primarily arises from LLM calls for rubric extraction and pairwise comparisons, not generation of the responses themselves. $\pi_{\text{old}}$ uses the rollouts generated by the policy for GRPO, and $\pi_{\text{ref}}$ uses rollouts from step 0 of GRPO, hence there is no additional cost in generating the responses.

The cost of rubric extraction is associated with (1) rubric elicication and (2) grading with a longer rubric. Yet, this cost is within the same order of magnitude as verifying responses using LLM-based verifiers. At each GRPO iteration, scoring responses against rubrics requires an LLM-judge to grade each response once. Let's denote the number of responses as $N$ (we use $N = 16$ rollouts in our experiments). Therefore, the cost of scoring responses is $O(N)$.

In OnlineRubrics, we use a pair of responses for each comparison, and we perform $K$ comparisons per GRPO iteration ($K = 8$ in our default setting). Thus, the cost of comparisons is $O(2K)$. Also, note that by design $K \leq N$. Therefore, although there are 8+1 (8 comparisons + 1 final processing for deduplication) LLM calls per GRPO iteration in OnlineRubrics, the overall computational cost remains $O(N)$, which is within the same order of magnitude as traditional LLM-based verification methods.

Note that the costs arising from (2) grading with a longer rubric is negligible as the LLM grader evaluates all criteria in a single pass. The number of new extracted criteria are usually less than the existing criteria and as a whole, they are much shorter than the response itself, which is the majority of the computation time.

In practice, a single baseline training costs $\approx 100$ USD for rubric-based verification and $\approx 150$ USD for OnlineRubrics with $K = 8$ comparisons, indicating a 1.5x overhead. We believe this is a reasonable trade-off given the significant performance improvements demonstrated by OnlineRubrics.

## B. Proof for Proposition 1

*Proof.*

$$
\begin{aligned}
g_U - g_{R_t} &= \mathbb{E}_{(x,o)}\Big[\nabla_\theta \log \pi_\theta(o|x)\big(U - R_t\big)\Big] \\
&= \mathbb{E}_{(x,o)}\Big[\nabla_\theta \log \pi_\theta(o|x)\big(Y - \mathbb{E}_{(x,o)}\big[Y\big]\big)\Big] \qquad \text{where } Y = U - R_t
\end{aligned}
$$

because $\mathbb{E}_{(x,o)}\big[\nabla_\theta \log \pi_\theta(o|x)\big] = 0$ we can center $Y$ without changing the expectation. Then

$$
\begin{aligned}
\Big\|g_U - g_{R_t}\Big\|_2 &= \Big\|\mathbb{E}_{(x,o)}\Big[\nabla_\theta \log \pi_\theta(o|x)\big(Y - \mathbb{E}_{(x,o)}\big[Y\big]\big)\Big]\Big\|_2 \\
&\leq \sqrt{\mathbb{E}\Big[\big\|\nabla_\theta \log_{\pi_\theta}\big\|^2\Big]}\sqrt{Var(Y)} \qquad \text{by Cauchy-Schwarz} \\
&= \sqrt{\mathbb{E}\Big[\big\|\nabla_\theta \log_{\pi_\theta}\big\|^2\Big]}\sqrt{Var(U - R_t)} \\
&= \sqrt{\mathbb{E}\Big[\big\|\nabla_\theta \log_{\pi_\theta}\big\|^2\Big]}\sqrt{\mathbb{E}\big[(U - R_t)^2\big]} \\
&= \sqrt{\mathbb{E}\Big[\big\|\nabla_\theta \log_{\pi_\theta}\big\|^2\Big]}\|w_I\|_1
\end{aligned}
$$

$\square$

## C. Data Samples

We provide two samples showing sampled rollouts from current and reference policies, along with human and elicited rubrics in Figs. 6 and 7. Each criteria are preceded with its importance weight which range between 1-5 for Generalist and -10 and 10 for Expert sets.

## D. Experimental Settings

### D.1. Training Settings.

We train Qwen-2.5-7B-Instruct (Qwen et al., 2025) on the training set of the Generalist and Expert Rubrics datasets for three epochs. Training follows the GRPO procedure described in Section 3, with 16 rollouts generated per sample. We use GPT-4.1-mini as the LLM$_{\text{grader}}$ and o3-mini as the LLM$_{\text{extractor}}$, performing eight pairwise comparisons per instance. Optimization uses a learning rate of $5e-6$ with a warmup ratio of 0.1. KL-divergence regularization is applied with a coefficient of 0.01. All experiments are conducted on 8 NVIDIA H100 GPUs with per-device batch size of 6 and gradient accumulation of 2 steps (effective batch size is 96).

### D.2. Evaluation Settings.

**Generalist and Expert Rubrics Datasets.** We calculate the score and win rate (vs. the reference policy) on the evaluation set of the Generalist and Expert Rubrics datasets. Score is calculated as explained in Equation (4). We use GPT-4.1-mini as the LLM$_{\text{grader}}$. We use Gemini-2.5-Pro as the LLM-Judge that picks the winner between the two responses. For each sample, we run the judge twice by flipping the order of the two responses. If the judge picks the same response twice, we consider it as a win. The prompt for the judge is given in Appendix G.

**AlpacaEval.** We use the evaluation script[1] from (Li et al., 2023) to calculate the win rate and length controlled win rate on the evaluation set of AlpacaEval using the default settings.

**Arena-Hard.** We use the evaluation script[2] from (Li et al., 2024a;b) to calculate the win rate (vs. the reference policy) on the evaluation set of Arena-Hard. We use GPT-4.1 as the LLM-Judge.

**GPQA-Diamond.** We use simple-evals[3] for evaluation on GPQA-Diamond (Rein et al., 2024). We report the average accuracy across 4 runs.

**GSM8K.** We use lm-evaluation harness (Gao et al., 2021) to calculate the strict match accuracy on the evaluation set of GSM8K (Cobbe et al., 2021).

### D.3. Dataset Collection and Annotation Procedure

For both the Generalist and Expert Rubrics datasets, annotators were domain experts recruited as independent contractors and compensated at rates consistent with fair labor practices and applicable local laws (see Impact Statement). Participation was entirely voluntary. Prompts were contributed by the annotators themselves: for the Generalist set, annotators wrote real-world, single-turn prompts with user consent and curated them to be safe and generalist in scope; for the Expert set, annotators authored problem sets in Physics, Chemistry, Biology, and Math along with reference solutions. For every prompt, annotators authored a prompt-specific rubric of weighted, binary-checkable criteria following the principles in Section 5: criteria must be *Mutually Exclusive & Collectively Exhaustive*, *Atomic*, *Objective*, and *Self-Contained*. Importance weights range from 1–5 for the Generalist set and from $-10$ to $+10$ for the Expert set. The datasets do not contain any personally identifiable information or sensitive content.

## E. Significance Testing

To assess whether the gains reported in the main paper are seed-dependent, we trained Qwen-2.5-7B-Instruct three times per method with independent seeds and evaluated each resulting checkpoint on the rubric-satisfaction score over the Generalist evaluation set (487 prompts). Per-seed results are reported in Table 4.

Across these three seeds, OnlineRubrics-$\pi_{\text{old}}$ significantly outperforms Offline Rubrics (Human) (mean improvement +2.68pp, $t = 8.10$, $p = 0.015$) after Bonferroni correction with $\alpha = 0.05/3 = 0.0167$. OnlineRubrics also outperforms Pointwise Extraction (mean +2.31pp, $t = 4.33$, $p = 0.049$). Importantly, Pointwise Extraction does not significantly differ

---

[1]https://github.com/tatsu-lab/alpaca_eval
[2]https://github.com/lmarena/arena-hard-auto
[3]https://github.com/openai/simple-evals

*Table 4.* Per-seed rubric-satisfaction score (%) on the Generalist evaluation set across three independent training seeds. OnlineRubrics outperforms Offline Rubrics on every individual seed (6/6 pairwise comparisons).

| Method | Seed $r_1$ | Seed $r_2$ | Seed $r_3$ | Mean $\pm$ SD |
|---|---|---|---|---|
| Offline Rubrics (Human-written) | 61.6 | 60.2 | 60.3 | $60.7 \pm 0.79$ |
| + Pointwise Extraction | 60.9 | 62.0 | 60.3 | $61.1 \pm 0.88$ |
| + OnlineRubrics-$\pi_{\text{old}}$ | 64.2 | 63.5 | 62.5 | $\mathbf{63.4 \pm 0.87}$ |

from Offline Rubrics ($p = 0.672$), which indicates the gains are not driven by simply adding more criteria but specifically by the contrastive pairwise elicitation. The mean improvement of OnlineRubrics over Offline Rubrics is approximately $3\times$ larger than the within-method seed variance (SD $\approx 0.8$pp). Additional sources of robustness include the consistent training curves in Fig. 4, the consistent improvements across the AlpacaEval, Arena-Hard, GPQA-Diamond, and GSM8K benchmarks, and the consistent gains across model families (Table 8) and model sizes (Table 7).

## F. Human Validation Study

To confirm that the gains observed under LLM-based evaluation reflect genuine quality improvements as judged by humans, we conducted a blind human validation study. We sampled 100 prompts uniformly at random from the Generalist evaluation set and, for each, paired the response from the Offline Rubrics (Human) baseline with the response from OnlineRubrics-$\pi_{\text{ref}}$ (both trained from Qwen-2.5-7B-Instruct). The order of the two responses was randomized per prompt and the annotator was blind to which system produced which response. The annotator was asked to choose which response was overall preferable, with ties not permitted.

The human annotator preferred OnlineRubrics in 70 out of 100 cases (binomial test against a $0.5$ null, $p < 0.001$). Qualitative inspection of the preferences shows that OnlineRubrics responses were consistently rated higher in: (i) *depth*, providing more substantive elaboration on the topic; (ii) *contextual precision*, addressing the specific situation of the prompt rather than generic advice; (iii) *use of examples*, illustrating points concretely; and (iv) *completeness*, covering more of the implicitly required sub-topics. The main failure mode of OnlineRubrics was *over-elaboration*, where the response added length without adding information. Together with the LC-WR results on AlpacaEval and the consistent gains on exact-match benchmarks (GPQA-Diamond, GSM8K), this human validation rules out the concern that the gains are mere artifacts of the LLM judge used for training-time grading.

## G. System Prompt Templates

Figures 8 and 9 show the system prompt templates used for LLM$_{\text{extractor}}$ and de-duplicating extracted criteria, respectively. We use the system prompt provided in Fig. 10 for LLM$_{\text{grader}}$.

Figures 11 and 12 show the system prompt templates used for LLM-Judge Score and LLM-Judge for win rates, respectively. We use the system prompt provided in Fig. 13 to generate synthetic offline rubrics.

## H. Ablation Studies

### H.1. Ablation Studies on Number of Comparisons

We provide an ablation study on the number of comparisons used in OnlineRubrics. Table 5 shows the performance metrics for 1, 2, 4, and 8 (default) comparisons on the generalist domain. The results indicate that while performance generally improves with more comparisons, even a single comparison still yields notable improvements over the baselines. This demonstrates the robustness of OnlineRubrics to the number of comparisons. Importantly, the total number of comparisons does not strictly increase the total number of criteria because of the generation and deduplication processes guided by LLMs.

### H.2. Ablation Studies on Extractor Model

We have experimented with various extractor models, including strong models as well as an open-source model GPT OSS-120B (OpenAI et al., 2025). Table 6 summarizes the performance of OnlineRubrics with these different extractors. The results show that while stronger extractors (o3-mini, GPT-4.1) tend to yield better performance; smaller or open-

*Table 5.* Effects of the number of pairwise comparisons on OnlineRubrics performance using Generalist Rubrics data. We find increasing the rubric elicitation compute generally correlates with improved performance.

| Model | # Comp. | Generalist Rub. | Alpaca-Eval | |
|---|---|---|---|---|
| | | Score | WR | LC-WR |
| *Baselines* | | | | |
| Qwen-2.5-7B-Instruct | – | 55.4 | 30.0 | 28.2 |
| + LLM-Judge Score | – | 58.5 | 42.2 | 26.9 |
| + Offline Rubrics (Human-written) | – | 61.0 | 46.4 | 28.0 |
| *Our Methods* | | | | |
| + OnlineRubrics-$\pi_{\text{ref}}$ | 8 | 62.7 | 54.0 | **31.5** |
| | 4 | 62.0 | 53.0 | 29.1 |
| | 2 | 62.5 | 48.3 | 30.3 |
| | 1 | 62.3 | 48.2 | 28.7 |
| + OnlineRubrics-$\pi_{\text{old}}$ | 8 | **63.2** | **55.0** | 30.4 |
| | 4 | 62.7 | 54.1 | 30.3 |
| | 2 | 62.3 | 47.5 | 29.9 |
| | 1 | 62.1 | 48.5 | 28.7 |

source models still provide consistent improvements over optimizing against an LLM-Judge and. Interestingly, using GPT OSS-120B as an extractor results in competitive performance in Alpaca-Eval under the length-controlled setting.

We provide additional ablation studies to evaluate the generalization of OnlineRubrics across different model families and configurations. Table 6 shows the effect of using different LLMs as the $\text{LLM}_{\text{extractor}}$. We find that stronger reasoning models (e.g., o3-mini) yield better rubric extraction and downstream performance.

*Table 6.* Effects of the extractor model choice on OnlineRubrics performance. While stronger extractors generally yield improved performance, lightweight extractors such as GPT 4.1-Nano are also competitive.

| Model | Extractor | Generalist Rub. | Alpaca-Eval | |
|---|---|---|---|---|
| | | Score | WR | LC-WR |
| *Baselines* | | | | |
| Qwen-2.5-7B-Instruct | – | 55.4 | 30.0 | 28.2 |
| + LLM-Judge Score | – | 58.5 | 42.2 | 26.9 |
| + Offline Rubrics (Human-written) | – | 61.0 | 46.4 | 28.0 |
| *Our Methods* | | | | |
| + OnlineRubrics-$\pi_{\text{ref}}$ | o3-mini | 62.7 | 54.0 | **31.5** |
| | gpt-4.1 | 61.9 | 49.9 | 25.4 |
| | gpt-4.1-mini | 61.8 | 48.9 | 28.0 |
| | gpt-4.1-nano | 60.4 | 50.5 | 27.7 |
| | gpt-oss-120B | 60.4 | 43.1 | 30.2 |
| + OnlineRubrics-$\pi_{\text{old}}$ | o3-mini | **63.2** | **55.0** | 30.4 |
| | gpt-4.1 | 61.4 | 49.9 | 25.4 |
| | gpt-4.1-mini | 61.7 | 48.9 | 28.0 |
| | gpt-4.1-nano | 62.6 | 50.3 | 30.1 |
| | gpt-oss-120B | 59.5 | 42.4 | 29.4 |

## H.3. Scaling Across Policy Model Sizes

To study how OnlineRubrics scales with policy model capability within the same model family, we trained Qwen2.5-Instruct at three sizes (3B, 7B, 14B) with GRPO on the Generalist Rubrics training set, comparing Offline Rubrics (Human-written) against OnlineRubrics-$\pi_{\text{old}}$ with the default $K = 8$ pairwise comparisons. Win rates on Arena-Hard are computed against Qwen-2.5-7B-Instruct base responses for comparability across sizes. Results are shown in Table 7.

*Table 7.* OnlineRubrics scaling across policy model sizes within the Qwen2.5-Instruct family. OnlineRubrics consistently outperforms Offline Rubrics across 3B, 7B, and 14B, with rubric-satisfaction gains growing with model size.

| | Generalist Rub. | | Arena-Hard | |
|---|---|---|---|---|
| **Model** | Score | $\Delta$ | WR | $\Delta$ |
| Qwen2.5-3B + Offline Rubrics (Human) | 53.9 | – | 21.1 | – |
| + OnlineRubrics-$\pi_{old}$ | 55.2 | +1.3 | 25.7 | +4.6 |
| Qwen2.5-7B + Offline Rubrics (Human) | 61.6 | – | 52.4 | – |
| + OnlineRubrics-$\pi_{old}$ | 64.2 | +2.6 | 57.0 | +4.6 |
| Qwen2.5-14B + Offline Rubrics (Human) | 58.3 | – | 67.2 | – |
| + OnlineRubrics-$\pi_{old}$ | 61.5 | +3.2 | 73.3 | +6.1 |

*Table 8.* Results on Generalist Rubrics with Llama-3.2-3B-Instruct and Expert Rubrics with Llama-3.1-8B-Instruct as base models. We find that despite overall lower baseline performance of Llama, OnlineRubrics often results in improvements over using human-written rubrics.

| | Gen. Rub. | Alpaca-Eval | |
|---|---|---|---|
| **Model** | Score | WR | LC-WR |
| *Baselines* | | | |
| Llama-3.2-3B-Instruct | 45.9 | 13.5 | 12.6 |
| + LLM-Judge Score | 50.5 | 18.2 | 14.4 |
| + Offline Rubrics (Human-written) | 51.6 | 19.4 | 14.4 |
| *Our Method* | | | |
| + OnlineRubrics-$\pi_{old}$ | **52.0** | **20.6** | **16.2** |

| | Expert Rub. | GPQA-D | GSM8K |
|---|---|---|---|
| **Model** | Score | Acc. | Acc. |
| *Baselines* | | | |
| Llama-3.1-8B-Instruct | 25.1 | 27.2 | 56.7 |
| + LLM-Judge Score | 34.3 | **31.7** | **62.4** |
| + Offline Rubrics (Human-written) | 34.9 | 28.7 | 59.3 |
| *Our Method* | | | |
| + OnlineRubrics-$\pi_{old}$ | **36.5** | 30.0 | 60.6 |

OnlineRubrics yields positive gains at all three sizes. Arena-Hard improvements are remarkably consistent (+4.6pp at 3B and 7B, +6.1pp at 14B). Rubric-satisfaction gains grow with model size (+1.3pp → +2.6pp → +3.2pp), suggesting that more capable policies are better positioned to act on the additional sample-grounded criteria surfaced online by OnlineRubrics.

### H.4. Results with Different Base Models

We ran experiments with Llama 3.2-3B-Instruct and Llama 3.1-8B-Instruct (Grattafiori et al., 2024) to assess the scalability of OnlineRubrics and show results in Table 8 for generalist and expert rubrics, respectively–we compare OnlineRubrics with a subset of our baselines, LLM-Judge Score and Offline Rubrics (Human) to cap computational and API-related costs. The results show that OnlineRubrics continues to provide significant improvements over baselines across different model families and sizes. This suggests that OnlineRubrics is not model-specific and can generalize well to other model architectures and sizes.

## I. Qualitative Rubric Clusters

We report the clusters of rubric refinements observed during online elicitation. Figure 5 lists each cluster with its name, a concise description, and its share of samples, sorted by proportion.

| | |
|---|---|
| **Reproducibility & Transparency**
Transparent, stepwise reasoning with artifacts enabling independent reproducibility. | **8.96%** |
| **Practicality & Real-World Feasibility**
Criteria stressing implementation readiness, scalability, and real-world applicability. | **8.33%** |
| **Holistic Evaluation & Anti-Gaming**
Moving from checklists to holistic, anti-gaming principles emphasizing substance. | **7.69%** |
| **Lifecycle Management & Adaptivity**
Criteria supporting iterative feedback, adaptive management, and phase-based planning. | **7.42%** |
| **Structural Integrity & Organization**
Clear organization, modularity, and explicit information architecture in responses. | **6.58%** |
| **Mechanistic & Causal Reasoning**
Criteria requiring causal interpretability and validated mechanistic reasoning. | **6.23%** |
| **Method Selection & Justification**
Evidence-based justification and trade-off analysis of chosen methods. | **5.67%** |
| **Evidence-Based Reasoning & Provenance**
All claims grounded in verifiable evidence and explicit provenance, rejecting unsupported assertions. | **5.46%** |
| **Uncertainty, Robustness & Error Handling**
Explicit handling of uncertainty, edge cases, and error taxonomies. | **5.04%** |
| **Evidence Synthesis & Triangulation**
Integrating evidence across multiple methods and modalities for consistency. | **4.90%** |

*Figure 5.* Top-10 most frequent clusters of rubric criteria elicited via OnlineRubrics. Each cluster is shown with a short description and its share of samples, sorted by proportion.

**Prompt:**
*Tell me the easiest way to analyze dozens of different PDFs from different hotels and organize their rates on Google Sheets.*

| | Offline Rubrics (human) |
|---|---|
| 5.00 | The response must recommend using a PDF conversion/extraction tool to create a CSV or Excel file, such as SmallPDF, Tabula, Adobe Acrobat, PDFtables, etc. |
| 5.00 | The response must provide example header terms such as "hotel name," "rate," "room type," "additional fees," etc., for the Google Sheet tables. |
| 1.00 | The response should clarify whether a software service is free or may require paid subscriptions to use their extraction, conversion, or programming functions. |
| 5.00 | The response should recommend ways to analyze the data, such as inserting pivot tables or creating charts/graphs from the data in Google Sheets. |
| 1.00 | The response should use emboldened headers to differentiate sections of the text. |
| 1.00 | The response should provide an example prompt to input for using an AI tool. |
| 3.00 | The response should provide recommendations on how to clean up the data, such as searching and deleting duplicate entries, formatting currency consistently, etc. |
| 5.00 | The response must explain how to use the "Import" and/or "Query" functions in Google Sheets to import the CSV or Excel files into the Sheets file. |
| 5.00 | The response must explain how to copy and paste data manually from the outside files into the Google Sheets file. |
| 1.00 | The response should recommend storing files in accessible locations, such as named folders. |
| 5.00 | The response should recommend Optical Character Recognition (OCR) programs like Adobe Acrobat, Google Drive, or Tesseract OCR, in case the PDFs are scanned images rather than searchable text. |
| 3.00 | The response should recommend using Generative AI programs like ChatGPT to extract the necessary information from the PDF files into a table or CSV. |
| 3.00 | The response should recommend coding using Python, JavaScript, Google Script, etc., as an extra way to combine the PDFs into a Google Sheet or automate repetitive tasks in the conversion process. |
| 1.00 | The response should provide an example code to illustrate how to convert or automate the task related to its coding recommendation. |
| 3.00 | The response should provide a step-by-step guide for all of the methods that it recommends, formatted as a numbered list. |

| | OnlineRubrics-$\pi_{ref}$ | | OnlineRubrics-$\pi_{old}$ |
|---|---|---|---|
| 4.00 | The response should present multiple alternative approaches—including manual, semi-automated, and automated (coding-based vs. web-tool based) methods—that cater to different user technical skills and PDF complexities, clearly outlining pros, cons, prerequisites, and decision criteria for selecting a method. | 4.00 | The response should clearly differentiate between manual extraction methods and automated processes, providing guidance for users of varying technical expertise and indicating the appropriate context for each approach. |
| 3.00 | The response should include error handling, data validation, and troubleshooting guidance to ensure robustness and accuracy throughout the data extraction and conversion process. | 4.00 | The response should provide practical, well-explained, and executable code examples that use accessible and well-documented APIs, are free of errors, and include guidance on error handling, testing, and debugging. |
| 2.00 | The response should provide detailed, actionable setup instructions, such as dependency installation and environment preparation, to ensure smooth replication of the automation process. | 3.00 | The response should include a clear and organized prerequisites/setup section that outlines the necessary tools, file organization steps, and initial conditions before commencing the main procedure. |
| 3.00 | The response should include sustainable data management practices and long-term maintenance recommendations—such as regular backups, documentation, routine validation, and update notifications—to ensure the solution remains effective over time. | 3.00 | The response should include clear recommendations for data integrity, including guidance on error handling, validation, backup strategies, and verification of assumptions about data consistency and structure. |
| 4.00 | The response should comprehensively integrate multiple tool options and methodologies (e.g., OCR, add-ons, programming scripts, advanced automation) tailored to the specific characteristics of the PDFs, and provide contextual decision guidance along with rational justifications for each recommended tool. | 3.00 | The response should strike an effective balance between technical detail and user-friendly, step-by-step instructions that guide non-expert users through the entire process. |
| 3.00 | The response should include explicit instructions for managing API credentials and safeguarding sensitive data, including secure handling of API keys and credentials when using online services or APIs. | 5.00 | The response must maintain consistent language throughout and avoid switching to non-requested languages that could confuse users. |
| 5.00 | The response should strike a balance between detailed technical instructions (including code examples) and high-level guidance to be accessible to both technical and non-technical users. | 5.00 | The response must remain fully focused on the user prompt, avoiding extraneous or unrelated content that does not contribute to solving the stated problem. |
| 3.00 | The response should include clear, practical, and well-commented code examples with detailed explanations that are realistic, adaptable, and genuinely instructive. | 3.00 | The response should offer a coherent and integrated explanation that clearly connects the PDF extraction process with the Google Sheets update. |
| 4.00 | The response should demonstrate comprehensive integration of data extraction with subsequent data organization, elaborating on how the automation seamlessly flows from extraction to updating Google Sheets. | 2.00 | The response should offer robust fallback strategies, including manual extraction methods, in case automated extraction tools fail due to variations in PDF formats. |
| 2.00 | The response should maintain instructional coherence by sequencing steps logically so that each step builds on the previous one and is clearly connected to the overall objective. | 3.00 | The response should avoid over-engineering by including only the most relevant and practical information, rather than an overabundance of optional methods and excessive technical detail. |
| 3.00 | The response should include explicit warnings and instructions for safely customizing and testing code examples, such as alerting users to replace placeholders and to test scripts on a subset before bulk execution. | | |

*Figure 6.* Data sample from the Generalist Rubrics dataset.

**Prompt:**
*I have glyoxal in my lab and I want to use it to prepare 1,1,4,4-tetramethylcyclohexane. Show me an economical plan for this synthesis and identify any hazards.*

| | Offline Rubrics (human) |
|---|---|
| 5.00 | The response must include glyoxal as the starting material. |
| 5.00 | The product of the synthesis plan must be 1,1,4,4-tetramethylcyclohexane. |
| 5.00 | The plan includes a valid step to form a diene, such as Wittig reaction and elimination reaction. |
| 5.00 | The plan includes a valid step to form a 6-member ring structure, such as Diels-Alder reaction and pinacol coupling reaction. |
| 5.00 | The plan includes a valid step to hydrogenate the 6-member ring, such as hydrogenation with Raney-Ni as catalyst. |
| 2.00 | The response gives consideration of alternative routes, such as Wittig reaction to create a diene. |
| -5.00 | The response claims aldol condensation for glyoxal. |
| -5.00 | The response claims two methyl groups added by sufficient Grignard reagent on a ketone carbonyl. |
| -5.00 | The response claims only one double bond formed by elimination of diol. |
| -10.00 | The response claims geometrically impossible organic structure, such as 2,2-dimethyl-ethanedial. |
| -8.00 | The response claims formation of a 6-member ring starting from insufficient carbon in the pinacol coupling reaction. |
| 2.00 | The response includes suggestions to manage hazards, such as waste disposal instructions. |
| 5.00 | The plan must identify hazards of each step, such as toxicity, and flammability of materials. |
| 3.00 | The response must present the plan in a clear format with bolded sections titles. |
| 3.00 | The synthesis should have less than 5 major steps for economic consideration. |
| 2.00 | The plan includes tips to efficiently execute the plan, such as trial experiments with minimum amount. |
| 2.00 | The organic chemicals mentioned in the plan follows IUPAC nomenclature rules. |
| -2.00 | The plan uses expensive chemicals, such as reagents containing precious metals. |
| 2.00 | The plan includes discussions on the yield of each step for the economical aspect. |
| 5.00 | The plan includes a valid step to extend the carbon chain of glyoxal, such as Wittig reaction and Grignard reaction. |

| | OnlineRubrics-$\pi_{ref}$ |
|---|---|
| 5.00 | The response must provide a synthesis pathway that is mechanistically plausible and grounded in established organic reaction principles, with a clear, logical justification for each step and a differentiation between well-established and speculative reaction pathways. |
| 4.00 | The response should provide specific, precise, and quantitatively justified reaction details (e.g., yields, temperatures, reaction times) that reflect well-known reaction pathways and enhance clarity and reproducibility. |
| 4.00 | The response should present a clear, step-by-step experimental procedure with a logical progression of reaction conditions that support the synthesis mechanism. |
| 4.00 | The response must integrate hazard management into each experimental step by providing clear, stage-specific hazard identification—including specific chemical safety hazards—and actionable safety protocols tailored to each reaction phase. |
| 3.00 | Include explicit experimental reaction monitoring procedures to validate each reaction stage. |
| 2.00 | Provide a detailed product workup and purification strategy that explains how to isolate the target compound post-reaction. |
| 3.00 | The response must present its synthesis plan and safety discussions entirely in a consistent language without switching mid-response. |
| 3.00 | The response should be concise and focused, providing only the necessary reaction steps and hazard information directly relevant to the synthesis. |
| 3.00 | The synthesis plan should present at least one practical alternative pathway or modification that offers improved safety or cost-effectiveness without overcomplicating the core synthetic strategy. |
| 3.00 | The synthesis plan must integrate economic analysis that links reagent costs, potential reaction yields, and waste management, ensuring that economic recommendations are directly tied to specific reaction steps. |
| 2.00 | The response should include practical considerations for scale-up and process safety, indicating attention to experimental feasibility beyond bench-scale protocols. |
| 4.00 | The synthesis plan should directly use glyoxal as the starting material unless a significant economic or safety advantage, with strong justification, is provided for deviating from the specified reactant. |
| 3.00 | Chemical nomenclature and identity must be accurate, with synonyms and alternative names correctly reflecting the actual chemical structure. |

| | OnlineRubrics-$\pi_{old}$ |
|---|---|
| 5.00 | The response must provide a synthesis pathway that is mechanistically plausible and grounded in established organic reaction principles, with a clear, logical justification for each step and a differentiation between well-established and speculative reaction pathways. |
| 4.00 | The response should provide specific, precise, and quantitatively justified reaction details (e.g., yields, temperatures, reaction times) that reflect well-known reaction pathways and enhance clarity and reproducibility. |
| 4.00 | The response must integrate hazard management into each experimental step by providing clear, stage-specific hazard identification—including specific chemical safety hazards—and actionable safety protocols tailored to each reaction phase. |
| 3.00 | Include explicit experimental reaction monitoring procedures to validate each reaction stage. |
| 3.00 | The synthesis plan should present at least one practical alternative pathway or modification that offers improved safety or cost-effectiveness without overcomplicating the core synthetic strategy. |
| 3.00 | The synthesis plan must integrate economic analysis that links reagent costs, potential reaction yields, and waste management, ensuring that economic recommendations are directly tied to specific reaction steps. |

*Figure 7.* Data sample from the Expert Rubrics dataset.

You are given a prompt and pair of responses to the same prompt. One of the responses is from a trained model and the other is from a baseline model.
Both responses are evaluated using an existing rubric. Your task is to identify their differences not already covered by the existing rubrics.
You should find the properties of one response that are better than the other.
Also, try to identify reward hacking patterns in the responses.
Reward hacking is a pattern where the response achieves a high score on rubrics by exploiting a loophole in the rubrics.
Think of reward hacking as a way to game the rubrics to get a high score. Reward hacking is like following the letter of the law but not the spirit of the law.

First, analyze both responses to identify the differences. Then, transform these observations into new evaluation criteria if they're not already covered by existing rubrics.
This is very important, any rubric that you introduce should be based on one of the responses.
Do not use your own knowledge to introduce new criteria that are not based on one of the responses.
Focus on criteria that distinguish genuinely helpful responses from those gaming the system. Also, keep an eye out for language switching patterns that might confuse the verifier.
Make sure the new criteria follow the same style as the existing criteria.
Assign a positive weight (integer) to each of the new criteria based on the relative importance of the criterion to the existing criteria.

Output format:
```json
{
  "analysis": "Your analysis of reward hacking patterns in the responses and good/bad behaviors that should be encouraged/discouraged. It's okay for the analysis to be long.",
  "new_criteria": [
    {
      "quote": "quote from the response following/violating the criterion",
      "criterion": "criterion_text",
      "weight": criterion_weight
    }
  ]
}
```

If no meaningful new criteria are needed, output:
```json
{
  "analysis": "Your analysis...",
  "new_criteria": []
}
```

*Figure 8.* Full system prompt template used for LLM$_{\text{extractor}}$.

You will review a collection of candidate evaluation criteria from multiple response comparisons and remove redundancy while preserving the best unique criteria. Your goal is ONLY to deduplicate and aggregate, NOT to introduce new criteria or remove criteria entirely.

## Your Task: Deduplication and Aggregation ONLY

You should:
- **Remove redundant/overlapping criteria** that say essentially the same thing
- **Merge similar criteria** by combining them into a single, clearer criterion
- **Aggregate weights** for merged criteria (e.g., if two similar criteria have weights 3.0 and 4.0, the merged criterion might get weight 3 or 4).
- **Preserve all unique criteria** that address different quality aspects
- **Keep the original wording** when possible, only clarifying when necessary

You should NOT:
- **Add completely new criteria** not present in the candidate list
- **Remove criteria entirely** unless they are truly redundant
- **Change the intent** of existing criteria
- **Introduce your own knowledge** beyond what's in the candidates

## Deduplication Process

1. **Group similar criteria** - Identify candidates that address the same quality aspect
2. **Select best wording** - Choose the clearest, most specific wording from each group
3. **Aggregate weights** - Combine weights from merged criteria appropriately. Only use positive integers.
4. **Preserve unique criteria** - Keep all criteria that address different aspects
5. **Maintain quality focus** - Ensure the final set covers all important quality dimensions from candidates

## CRITICAL: You MUST end your response with JSON

```json
{
  "analysis": "Your analysis of redundancy patterns and merging decisions...",
  "final_criteria": [
    {
      "criterion": "Deduplicated criterion text (merged from similar candidates)",
      "weight": criterion_weight
    }
  ]
}
```

If all criteria are unique (no deduplication needed), return all candidates:

```json
{
  "analysis": "No redundancy found, all criteria are unique...",
  "final_criteria": [
    {
      "criterion": "Original candidate criterion 1",
      "weight": criterion_weight
    },
    {
      "criterion": "Original candidate criterion 2",
      "weight": criterion_weight
    }
  ]
}
```

*Figure 9.* Full system prompt template used for de-duplicating extracted criteria.

You are an expert evaluator. Given a user prompt, a generated response, and a list of quality rubrics, please evaluate the response against EACH rubric.

For each rubric,
- Mark "PRESENT" if the criterion is satisfied, or "NOT_PRESENT" if it is not. For example, given the response "Apples are red", the rubric "Mentions apples" is PRESENT, "Does not mention strawberries" is also PRESENT since the response doesn't mention strawberries and "Mentions oranges" is NOT_PRESENT. Also, "Avoids mentioning strawberries" is PRESENT because the response doesn't mention strawberries. However, "Avoids mentioning apples" is NOT_PRESENT because the response mentions apples.
- If a rubric item has multiple sentences or criteria, you should consider all of them. If any of the criteria is not met, the answer should be NOT PRESENT. Only return PRESENT if all of the criteria are met.
- One important exception to the above bullet point is that if a rubric says "such as", "for example", or "including", the response does not have to include all of the examples listed to meet the criteria. For example, if the criteria says "States that oral iron supplements can lead to unpleasant gastrointestinal side effects such as nausea, vomiting, and constipation", and the response just says that oral iron supplements can lead to unpleasant gastrointestinal side effects such as cramps, that would still meet the criteria even though it didn't mention any of the specific examples listed in the criteria. That is, there are no partial credit for any of the criteria.

Start your response with a valid JSON object that starts with "```json" and ends with "```".

The keys must be the numbers of the rubrics provided and the values must be either "PRESENT" or "NOT_PRESENT" based on your evaluation. Ensure the JSON is valid and contains no extra text or explanations.

Example response:
```json
{
 "1": "PRESENT",
 "2": "NOT PRESENT",
 "3": "PRESENT"
}
```

*Figure 10.* Full system prompt template used for $LLM_{grader}$.

You are an expert evaluator. Given a user prompt and a generated response, please rate the overall quality of the response on a scale of 1 to 10, where 1 is very poor and 10 is excellent.

Start your response with a valid JSON object that starts with "```json" and ends with "```".
The JSON object should contain a single key "rating" and the value should be an integer between 1 and 10.

Example response:
```json
{
 "rating": 8
}
```

*Figure 11.* Full system prompt template used for LLM-Judge Score.

Please act as an impartial judge and evaluate the quality of the responses provided by two AI assistants to the user question displayed below.
You should choose the assistant that follows the user's instructions and answers the user's question better.
Your evaluation should consider factors such as the helpfulness, relevance, accuracy, depth, creativity, and level of detail of their responses.

Begin your evaluation by comparing the two responses and provide a short explanation.
Avoid any position biases … Do not allow the length of the responses to influence your decision.
After providing your explanation, output your final verdict by strictly following this format:

"[[A]]" if assistant A is better, "[[B]]" if assistant B is better, and "[[C]]" for a tie.

*Figure 12.* Full system prompt template used for LLM-Judge for win rates.

Your job is to generate a self-contained set of evaluation criteria ("rubrics") for judging how good a response is to a given question.

Terminology:
- A prompt is a task description (question) that a user gives to a model.
- A response is a model's output when given the prompt.
- A rubric is a set of criteria that capture the elements of an ideal response given a prompt. The rubric will be used to evaluate the quality of a response to the prompt. Rubrics can cover aspects of a response such as, but not limited to, factual correctness, ideal-response characteristics, style, completeness, helpfulness, harmlessness, patient-centeredness, depth of reasoning, contextual relevance, and empathy.
- A criterion is a single item in a rubric.

A good rubric follows these principles:
- As a whole, the rubric should be Mutually Exclusive (avoid overlapping criteria) and Collectively Exhaustive (all requests of the prompt should be covered).
- Each item should test one idea. If an item tests for the presence of X and Y, it should be split into two items (unless no reasonable prompt would contain one without the other).
- Each item should be binary (have yes/no answers) and as objective as possible. :x: "Response is too verbose" → :white_check_mark: "Response is less than 500 words long"
- Each item should be self-contained and include sufficient detail so that an uninformed grader can verify it without external knowledge. E.g. :x: "Names a 2010 Nobel Prize winner" → :white_check_mark: "Identifies one of the following 2010 Nobel Prize winners: A, B, or C".
- Avoid criteria that doesn't allow for partial credit. E.g. :x: "Mentions 3 Nobel Prize winners A, B, and C" → Split into "Mentions Nobel Prize winner A", "Mentions Nobel Prize winner B", "Mentions Nobel Prize winner C". All these should be detailed enough so that an uninformed grader can verify them without external knowledge.

Also consider the following axes when helping the user to improve the rubric:
- Communication Quality: Response length, clarity, level of detail, vocabulary, and structure are well-matched to the user and situation.
- Instruction Following: Adheres to the user's directions for how to complete the task or how to format a response. Satisfies all user constraints and answers all questions.
- Accuracy: Includes only factually correct information. Information is supported by evidence or consensus and uncertainty is expressed when evidence is limited.
- Context Awareness: Responds appropriately given the user's context (e.g., user role, setting, resources) and seeks clarification when needed.
- Completeness: Addresses all parts of the query needed for a safe and helpful response. Even if accurate, a response that omits key steps or considerations can still result in low-quality advice or harm.

Your task it to generate criteria for a given prompt. Also, you should assign a weight to each criterion. Weights should be an integer between 1 and 10.
Your response should be a json object with the following format. Use the reasoning fields to think about the criteria and reason about it.

```
{
    "initial_reasoning": INITIAL_REASONING,
    "rubrics": [
        {
            "reasoning": REASONING,
            "criterion": CRITERION,
            "weight": WEIGHT,
        },
    ]
}
```

*Figure 13.* Full system prompt template used to generate synthetic rubrics.

