# OpenReview forum: "Online Rubrics Elicitation from Pairwise Comparisons"
_ICML.cc/2026/Conference — ICML 2026 regular_

### Official Review · Reviewer_RyW3 · 2026-02-23

**Soundness:** 3
**Presentation:** 3
**Significance:** 3
**Originality:** 3
**Overall Recommendation:** 5
**Confidence:** 5

**Summary:**

A key limitation in rubric-based RL for LLM post-training is identified: they tend to be incomplete and can not adapt to emergent behaviors. In order to address this limitation, this paper proposed OnlineRubrics, a framework that generates per-prompt evaluation metric dynamicly during training.

Two datasets are constructed (Generalist and Expert Rubrics). The approach is also evaluated on benhcmarks datasets such as AlpacaEval, Arena-Hard, GPQA and GSM8K. The experiment results proved that OnlineRubrics outperforms static human or synthetic rubrics.

**Compliance With Llm Reviewing Policy:**

Affirmed.

**Final Justification:**

I think this is a very good paper that solves the key limitation in rubric-based RL for LLM post-training. My concern is fully solved by the author's rebuttal. It's hard for me to say that this paper has **exceptional impact** for strong acceptance, but it is perfect enough to be accepted.

**Key Questions For Authors:**

Does the performance of this method related to model size?

How about the robustness of this method? Is the performance sensitive to different randomness?

**Limitations:**

Please see the weaknesses part upward.

**Strengths And Weaknesses:**

## Strengths
1. This paper is well-written and easy to understand.

2. This paper presents a novel and promising approach to mitigate the limitations in existing Rubrics-based LLM post-training methods.

3. The proposed algorithm is clear and has sufficient mathematical support, making it easy to interpret.

4. This paper provided enough experimental results to support the approach they proposed. The experiments are conducted on diverse settings and datasets, covering Qwen and Llama families.

## Weaknesses
I think some experiments on different model sizes within one LLM family can make this paper better.

---

> ### Author Rebuttal · Authors · 2026-03-31
>
> Thanks for your thoughtful review. We're encouraged that you found OnlineRubrics to be novel, well-resented, and with enough experimental results. Please find our response to your concerns below:
>
> ### **W1 and Q1: Relation to Model Size**
>
> Thanks for your suggestion. We ran additional experiments within the Qwen2.5-Instruct family across three model sizes (3B, 7B, 14B) trained with GRPO on the Generalist dataset. Results are shown below (Arena-Hard win rates are against Qwen2.5-7B-Instruct base responses):
>
> | Model | Rubric Sat. | Δ | Arena-Hard WR | Δ |
> |---|---|---|---|---|
> | Qwen2.5-3B + Offline Rubrics (human-written) | 53.9% | — | 21.1% | — |
> |      + OnlineRubrics ($\pi_{old}$) | **55.2%** | +1.3pp | **25.7%** | +4.6pp |
> | Qwen2.5-7B+  Offline Rubrics (human-written)  | 61.6% | — | 52.4% | — |
> |      + OnlineRubrics ($\pi_{old}$) | **64.2%** | +2.6pp | **57.0%** | +4.6pp |
> | Qwen2.5-14B + Offline Rubrics (human-written) | 58.3% | — | 67.2% | — |
> |      + OnlineRubrics ($\pi_{old}$) | **61.5%** | +3.2pp | **73.3%** | +6.1pp |
>
> OnlineRubrics consistently outperforms the offline baseline at all three scales across all benchmarks. Arena-Hard gains are notably consistent (+4.6pp at 3B and 7B, +6.1pp at 14B), suggesting the pairwise elicitation generalizes well across model sizes. Rubric satisfaction improvements also grow with scale (+1.3pp → +2.6pp → +3.2pp), suggesting that larger models may better leverage the elicited criteria. This is an encouraging finding, as it suggests OnlineRubrics becomes more effective as model capability increases, which aligns with our intuition that a stronger policy is better positioned to leverage the additional signal provided by online elicited criteria. We will include these results in the final version.
>
> ### **Q2: Robustness and Randomness**
>
> We ran three independent training seeds per method and conducted cross-seed t-tests on rubric satisfaction scores across 487  generalist prompts:
>
> | Method | Seed r1 | Seed r2 | Seed r3 | Mean ± SD |
> |---|---|---|---|---|
> | Offline Rubrics (Human-written) | 61.6% | 60.2% | 60.3% | 60.7% ± 0.79pp |
> | +  Pointwise Extraction | 60.9% | 62.0% | 60.3% | 61.1% ± 0.88pp |
> | + OnlineRubrics ($\pi_{old}$) | 64.2% | 63.5% | 62.5% | 63.4% ± 0.87pp |
>
> ​​OnlineRubrics significantly outperforms the Offline Rubrics (mean +2.68pp, t=8.10, p=0.015) after Bonferroni correction (α=0.0167), and outperforms Pointwise Extraction (mean +2.31pp, t=4.33, p=0.049). Importantly, Pointwise Extraction does not significantly differ from offline rubrics (p=0.672). *OnlineRubrics outperforms the offline rubrics (human-written) in every single seed* (6/6 pairwise comparisons), and the mean improvement is approximately 3× larger than the within-method seed variance.
>
> We would also like to point to several additional sources of evidence that suggest the results are robust. First, the training curves in Figure 4 demonstrate consistent improvements of OnlineRubrics over baselines throughout the entire training process, suggesting the gains are stable rather than due to random chance. Second, OnlineRubrics consistently outperforms all baselines across multiple benchmarks (AlpacaEval, Arena-Hard, GPQA-Diamond, and GSM8K) spanning both instruction-following and reasoning tasks. Third, our ablations in Table 6 show that OnlineRubrics yields consistent improvements across different base model families (Qwen2.5-7B, Llama 3.2-3B, Llama 3.1-8B), further supporting the robustness of the method.

---

> > ### Author Rebuttal · Reviewer_RyW3 · 2026-04-02
> >
> > Thanks for adding these experiments. My concern is addressed, and I will raise my confidence for 5 Accept.

---

### Official Review · Reviewer_dUgz · 2026-03-11

**Soundness:** 2
**Presentation:** 3
**Significance:** 3
**Originality:** 4
**Overall Recommendation:** 5
**Confidence:** 3

**Summary:**

This paper proposes OnlineRubrics, a framework that dynamically elicits additional rubric criteria during RL training by comparing pairs of responses, rather than relying only on a fixed human-written rubric. The main idea is that static rubrics can miss emerging failure modes or desired behaviors as training progresses, especially for open-ended tasks. The method augments existing rubrics online and uses the resulting rubric-based reward during training. Empirically, the paper shows gains over several baselines, including LLM-judge scoring, synthetic offline rubrics, human-written offline rubrics, universal requirements, and pointwise elicitation. The reported improvements are consistent across both generalist and expert settings, as well as several external benchmarks.

**Compliance With Llm Reviewing Policy:**

Affirmed.

**Final Justification:**

My main concerns are addressed in the rebuttal.

**Key Questions For Authors:**

See weakness.

**Limitations:**

yes

**Strengths And Weaknesses:**

## Strengths
1. The paper addresses an important and timely problem. The motivation is strong because fixed rubrics are often incomplete and may become less reliable as model behavior changes during RL training.
2. The method is simple and intuitive. Using **pairwise comparison** to surface missing criteria is a natural design choice, and the comparison against pointwise elicitation supports the claim that response contrast helps discover better criteria.
3. The empirical evaluation is reasonably broad. The authors test both generalist and expert settings, include several external benchmarks, and compare against a meaningful set of baselines. The gains appear fairly consistent.


## Weaknesses
1. The method still relies heavily on **LLM-generated supervision to improve LLM-generated supervision**. It seems possible that the major improvement comes from small model distill knowledge from larger model via these rubrics. Could the authors clarify to what extent the observed performance gains are actually driven by knowledge distillation from the larger extractor model via these newly generated rubrics?
2. The training setup is not fully clear from the current draft. The paper states that it uses GRPO as the training algorithm, and Section 6.2 presents baselines such as LLM-Judge Score and Offline Rubrics as different reward constructions. However, it is not explicitly stated whether all of these baselines are also trained GRPO. If not, then it needs to add a baseline of using GRPO alone.
3. Theoretical analysis is weak. Proposition 1 seems more like a heuristic explanation rather than a rigorous proof.
4. New rubric criteria are elicited by an LLM, and another LLM grader is used to score responses. The paper does not fully establish whether the learned criteria are truly better aligned with human judgment, rather than simply better matched to the chosen evaluator. If author claims interpretability, then human validation is needed.

---

> ### Author Rebuttal · Authors · 2026-03-31
>
> Thanks for your thoughtful review. We're glad that you found our work timely, intuitive, and empirical evaluation broad and meaningful. Please find our responses to your concerns below:
>
> ### **W1: Reliance on LLMs in Extraction**
>
> We would like to clarify that the LLM_{Extractor} is explicitly instructed not to use its own knowledge to introduce new criteria (Figure 8): "Do not use your own knowledge to introduce new criteria that are not based on one of the responses." We believe this should substantially limit the extent to which the gains can be attributed to knowledge distillation from the larger extractor model, since extractor focuses on differences already present in the rollouts, not as a source of new knowledge. We will add this to the limitations section in the final version. We also ran an ablation on different extractor models in Section 6.4, and found that while the choice of the extractor model does affect performance gains obtained with OnlineRubrics, even lightweight models such as GPT4.1Nano are competitive with o3-mini.
>
> ### **W2: Training Setup**
>
> Section 6 (Experiments and Results) states that “We train Qwen-2.5-7B-Instruct *with GRPO as the training algorithm* on the train set from both Generalist and Expert Rubrics datasets.” We also describe the GRPO setup in Section 3.1.
>
> To clarify, all baselines use the same training algorithm (GRPO) and differ only in how the reward is constructed. Thus, there is no setting where GRPO is used “alone,” as it always requires a reward function.
>
> Specifically, the “LLM-Judge Score” baseline uses an LLM judge to assign rewards and the offline rubric baselines use fixed human or synthetic rubrics. We show that OnlineRubrics outperforms all the baselines.
>
> ### **W3: Theoretical Analysis**
>
> We clarify that this section is not intended to provide asymptotically tight RL optimization guarantees, rather it formalizes why eliciting missing criteria matters in rubric-based reward modeling. We note that because the dependency on missing criteria mass $||w_I||_1$ is unavoidable, missing latent criteria induces error in reward calculation which in turn induces gradient misalignment. We will revise the presentation of this section to make this intention more explicit.
>
> ### **W4: LLM and Human Evaluation Alignment**
>
> We appreciate this concern and partially agree. However, we would like to highlight several design choices that mitigate this concern. First, our evaluation on the curated Generalist and Expert sets uses human-written rubrics with no elicited criteria. Second, we use a different model as the LLM-Judge for win rate evaluation (Gemini-2.5-Pro) than the one used for grading during training (GPT-4.1-mini), reducing the risk that improvements simply reflect better alignment with a single evaluator's preferences. Also, our out-of-distribution evaluations, particularly GSM8K and GPQA-Diamond which use exact-match accuracy, showing consistent improvements.
>
> Following your suggestion, we conducted a blind human validation study comparing OnlineRubrics ($\pi_{old}$) and Human rubrics baseline on 100 randomly sampled prompts. Human annotator preferred OnlineRubrics in 70 out of 100 cases (binomial test p < 0.001), with qualitative analysis showing that OnlineRubrics responses were consistently better in depth, contextual precision, providing examples, and completeness. The main failure mode was over-elaboration, where extra length was unnecessary. These results confirm that the improvements are not evaluator-specific artifacts but reflect genuine quality improvements as judged by humans. We will include this human validation study in the final version.

---

> > ### Author Rebuttal · Reviewer_dUgz · 2026-04-03
> >
> > My main concerns are addressed in the rebuttal. I raised by score to 5.

---

### Official Review · Reviewer_ymJa · 2026-03-12

**Soundness:** 3
**Presentation:** 3
**Significance:** 3
**Originality:** 3
**Overall Recommendation:** 5
**Confidence:** 4

**Summary:**

In this paper the authors introduce a technique called OnlineRubrics.

Motivation:
- With the new paradigm of rubrics as reward, offline rubrics generally have the following issues: (a) they don’t capture the nuances of mistakes or errors that happen in responses during training (b) models can have reward hacking behavior where they add text that games the LLM as a judge.

Method:
During the training, at each step they do a certain number of pairwise comparisons (between the current policy and a control policy - either a reference model or a previous iteration policy) and extract criteria in an online manner, in addition to offline criteria. Then they update the criteria list, generate scores and use that as reward

Results:
- They show that this way of training beats (a) LLM as a judge scores (b) use of synthetic rubrics (c) use of human rubrics (d) pointwise criteria generation
- They show training results on Qwen-2.5-7B-Instruct and Llama-3.2-3B-Instruct
- They show that this technique works with some open-source evaluators like GPT-OSS as well
- The choice of pairwise comparisons they have are optimal, anything less than that would lead to sub par results.
- They have a qualitative analysis showing how the online criteria generated might capture reward hacking behavior

**Compliance With Llm Reviewing Policy:**

Affirmed.

**Final Justification:**

the authors addressed my questions in the rebuttal phase, they showed the differences are meaningful between using different types of rubrics and their approach does better than using offline rubrics or human rubrics (over different seeds/runs). They also addressed the past work and limitation concerns.

**Key Questions For Authors:**

1. Are the differences reported in the results table significant? Please report/indicate this. If not, then why?
2. Do you have any insights into why the expert datasets performance is much lower than general datasets? Can it be attributed to any fundamental limitation of the method? Or is it about the type of rubrics that are distilled from the comparison?
3. On average how many rubrics do the different methods generate? How does the number change as the training progresses for the proposed method? And is there a shift in the specificity/focus of these as the training progresses? can you give examples?
4. How does your work compare to the existing work which are missing as references [mentioned in weaknesses]

It’ll be nice to have clarification on the following:
In the method, you let the LLM extractor decide the weight of the rubrics, did you validate if this weight makes sense? Are there certain kinds of rubrics that consistently weighted positive vs negative (maybe indicating that the model is making a certain kind of mistake consistently?)

Might be good to include more details in the paper regarding the following [doesn’t change score]:
1. Can you give more information on how the two datasets were collected? where the questions were sourced from, how did you hire the annotators, what was their expertise, exact annotation instructions, compensation etc?

**Limitations:**

yes

**Strengths And Weaknesses:**

Strengths:
1. The paper proposes a very relevant and timely method. Using online rubrics as reward signals for LLMs is an important direction.
2. Their empirical evaluation is quite rigorous. The authors test across multiple datasets, and base models. Use appropriate LLM-as-a-judge comparisons, taking into account the cost vs performance trade offs. They include competitive baselines.
3. Their ablations are thorough as well. They ablate the choice of the extractor and the number of pairwise comparisons necessary for their system to work.
4. They include a discussion on the computational costs and share all prompts which makes the research transparent and reproducible.

Weaknesses:
1. The paper lacks any significance testing reporting on key comparisons. This is useful for knowing if any difference reported as improvement is meaningful or noise.
2. The contribution of online rubric is not super well isolated. We see improvements between LLM as a judge score and all rubric baselines, but the difference between using human  {offline} vs pointwise {online} vs pairwise {online} is less clear.
3. The paper is missing a limitations section that might be helpful in understanding where the ceiling for this method is.  (a) For example, given a performance gap between expert and general datasets, is that something to do with the rubrics captured or any signal that was unable to be derived from the rubrics. (b) Are there any limitations on using LLM-as-a-judge for rubric and reward generation?
4. Two major relevant works are missing from the related literature and discussion:
(a) OpenRubrics: Towards Scalable Synthetic Rubric Generation for Reward Modeling and LLM Alignment
(b) DR Tulu: Reinforcement Learning with Evolving Rubrics for Deep Research
Both of these leverage the idea of using contrastive rubric generation for training models.

---

> ### Author Rebuttal · Authors · 2026-03-31
>
> Thanks for your thoughtful review.
>
> ### **W1, W2, and Q1: Significance Testing**
>
> We ran three independent trainings per method and conducted cross-seed t-tests on rubric satisfaction scores across generalist prompts:
>
> | | r1 | r2 | r3 | Mean ± SD |
> |---|---|---|---|---|
> | Offline Rubrics (Human) | 61.6% | 60.2% | 60.3% | 60.7% ± 0.79pp |
> |   + Pointwise Extraction | 60.9% | 62.0% | 60.3% | 61.1% ± 0.88pp |
> |   + OnlineRubrics (pi_old) | 64.2% | 63.5% | 62.5% | 63.4% ± 0.87pp |
>
> OnlineRubrics significantly outperforms the human-written offline rubrics (mean +2.68pp, t=8.10, p=0.015) after Bonferroni correction (α=0.0167), and outperforms Pointwise Extraction (mean +2.31pp, t=4.33, p=0.049). Pointwise Extraction does not significantly differ from offline rubrics (p=0.672). *OnlineRubrics outperforms the offline rubrics in every single seed (6/6 pairwise comparisons). This shows that the improvement is not simply due to having more rubrics, but specifically due to the contrastive pairwise comparison.
>
> ### **W3: Limitations**
>
> We will add the following discussion:
> > Our proposed technique is reliant on LLM generations which are prone to errors and biases which may propagate into the generated criteria. Similarly, although common practice, our evaluations on the curated eval set, AlpacaEval and Arena-Hard are also LLM-based and are subject to LLM errors. Finally, when the most recent policy is used as the control model, rubrics may undergo semantic drift; while we hypothesize that using the reference policy as the control can reduce this drift, future work can directly measure this effect.
>
> ### **W4 and Q4: Related Work**
>
> Thanks for pointing us to these works. We will add the following:
>
> > Related work has proposed generating synthetic rubrics from instructions and using them to train rubric-conditioned reward models in an offline setting [a]. Unlike them, we elicit rubrics in an online matter directly from pairwise comparisons instead of instructions.
>
> > Concurrent work on generating rubrics in an online matter have attempted to generate rubrics from model rollouts in a free-form manner while having access to external knowledge [b]. OnlineRubrics, on the other hand, elicits new criteria from pairwise comparisons vs. a control policy instead of an external knowledge source. This results in more grounded rubrics focusing on emerging behaviors during training.
>
> We would like to also highlight that Dr Tulu [b] cites our work as a concurrent work, stating that we “explore generating online rubrics by contrasting pairwise or multiple model rollouts in a closed-book setting.“
>
> ### **Q2: Expert Performance and Method Limitations**
>
> OnlineRubrics, by design, elicits criteria that are grounded in one of the two responses being compared. In generalist domains, at least one response typically exhibits a clear desired or undesired behavior. On the other hand, for the expert domain, given that we’re working with a relatively weak policy model, often the correct reasoning or domain-specific insight is absent from both responses. Therefore, the elicitation process has no basis to surface it. The method's ceiling is therefore bounded by the quality of the rollouts themselves.
>
> ### **Q3: Extracted Criteria Analysis**
>
> As stated in Section 4.1, OnlineRubrics extracts approximately eight criteria per sample by default. We do observe a slight growth in the number of elicited criteria as training progresses. However, rubrics do not accumulate over epochs, so there is no risk of unbounded growth.
>
> Regarding the shift in focus, we already analyze this in Section 6.5: over the course of training, elicited criteria tend to move beyond surface-level correctness toward meta-criteria such as structural organization, causal reasoning, and anti-gaming principles. We have also included concrete examples in both sections.
>
> ### **Q: Clarification on weights**
>
> The elicited weights are anchored in two ways: First, the extractor is provided with the existing human rubrics (and their weights) as a reference. Second, the deduplication process is guided to reflect how frequently a criterion was raised across multiple pairwise comparisons. We manually inspected 50 samples and found their weights to be sensible.
>
> Regarding positive vs. negative weights, we note that this is largely a matter of phrasing rather than a fundamental distinction. For example, "does not use unsafe language" (+5) is equivalent to "uses unsafe language" (-5). In practice, we found that the assigned weights align with the intended direction of the criterion.
>
> ### **Q: Annotation**
>
> The annotators for both datasets were domain experts recruited as independent contractors and compensated fairly, as stated in the Impact Statement. For the both datasets, prompts were contributed by annotators, with rubrics created following the principles outlined in Section 5: criteria must be Atomic, Descriptive, and Objective. We will add detailed instructions to the final version.

---

> > ### Author Rebuttal · Reviewer_ymJa · 2026-04-03
> >
> > Thanks for the response! My concern is addressed, and I will raise my score to 5.

---

### Official Review · Reviewer_vfiR · 2026-04-03

**Soundness:** 3
**Presentation:** 2
**Significance:** 2
**Originality:** 3
**Overall Recommendation:** 3
**Confidence:** 4

**Summary:**

This paper addresses the limitations of static rubrics in LLM reinforcement learning—such as vulnerability to reward hacking and inability to capture emerging features during training—by proposing OnlineRubrics. This method dynamically generates new evaluation criteria during training through pairwise comparisons between the outputs of the current policy model and a reference model (or an older policy). Experimental results show that OnlineRubrics improves performance by up to 8% over using only manually crafted static rubrics and by 25% over the baseline on benchmarks such as AlpacaEval, GPQA, and ArenaHard.

**Compliance With Llm Reviewing Policy:**

Affirmed.

**Final Justification:**

In the traditional RLHF paradigm, Rubrics for long-text generation are often offline and static. This paper acutely captures the pain point that static Rubrics are prone to Reward Hacking. Although the concept of Online Rubric is interesting, its underlying technology still heavily relies on closed-source powerful LLMs (such as o3-mini / GPT-4.1) as Extractor and Grader. I will maintain my judgment.

**Key Questions For Authors:**

see the weaknesses

**Limitations:**

no. (1) Lack of comparison with strong baselines in Online RL; (2) Although the concept of Online Rubric is intriguing, its underlying technology still heavily relies on the assembly of pipelines that utilize powerful closed-source LLMs (such as o3-mini / GPT-4.1) as Extractors and Graders

**Strengths And Weaknesses:**

Strengths:
1.	The experimental results are sufficient
2.	The performance seems good.
Weaknesses
Weakness 1: Insufficient motivation for cross-strategy comparison
Although the authors’ critique of static scoring criteria is well-founded, the derivation from the “limitations of static scoring” to the “need for pairwise comparison between $P_{curr}$ and $P_{ref}$” lacks sufficient logical and empirical support.
Core issue: The paper does not explain why the diversity required for scoring criteria extraction must come from cross-strategy comparison. If the goal is to extract criteria that distinguish high-quality responses from low-quality ones, why not directly use multiple samples from the current policy itself (i.e., $P_{curr}$ vs. $P_{curr}$)?
Missing ablation studies: The paper clearly lacks crucial ablation experiments. The authors should provide quantitative comparisons for:
1.	$P_{curr}$ vs. $P_{ref}$
2.	$P_{curr}$ vs. $P_{curr}$
3.	Comparisons involving randomly diversified samples
Without these comparisons, it is impossible to demonstrate that $P_{ref}$ provides meaningful “optimization directions” rather than merely highlighting irrelevant distributional variations. Furthermore, there is no intuitive explanation of why this method works.

Weakness 2: May capture style drift rather than fundamental capability
Assuming that the differences captured through pairwise comparisons are inherently useful for generating new scoring criteria is problematic. These differences often reflect superficial “style drift” (e.g., fluctuations in response length, tone, or formatting) rather than genuine evolution of the model’s underlying reasoning or task-solving abilities.
Risk of random walk: If the scoring criteria extractor (Extractor LLM) produces or over-interprets criteria such as “conciseness” or “tone” due to incidental differences between two samples, the reward function may fall into a “random walk.” This can destabilize the training signal, causing the model to oscillate between superficial features without achieving real performance improvement. The authors need to show how the system distinguishes spurious style noise from key evaluation criteria.

Weakness 3: Lack of background and comparison with prior work
The paper fails to adequately situate itself within the broader context of dynamic scoring criteria generation. To assess the novelty and effectiveness of the proposed approach, it is necessary to provide empirical or analytical comparisons with existing state-of-the-art methods.
Specific baseline comparison: For example, the authors should compare their method qualitatively and quantitatively with “QuRL: Rubrics As Judge For Open-Ended Question Answering.”

Weakness 4: Introduction fails to reach a clear conclusion
The introduction does not provide a coherent logical flow and lacks a summary of contributions, making it even harder for readers to grasp the key points.

---

### Decision · Program_Chairs · 2026-04-30

**Decision:**

Accept (regular)

**Comment:**

Summary: This paper proposes OnlineRubrics, a method that dynamically elicits evaluation criteria during RL-based LLM post-training by performing pairwise comparisons between the current policy and a reference policy, addressing the limitations of static rubrics that are vulnerable to reward hacking and fail to capture emergent behaviors. Results are evaluated on three benchmarks for instruction following.

Strengths: The method is well-motivated and addresses a timely problem — static rubrics becoming stale during training. The empirical evaluation is rigorous: multi-seed significance testing confirms OnlineRubrics significantly outperforms offline rubrics, with gains consistent across model scales, base model families, and external benchmarks. A human validation study confirms improvements are not evaluator-specific artifacts.

Weakness: The method relies on proprietary closed-source models at every stage of the pipeline and evaluates primarily with closed-source models. This is a practical limitation that was not fully addressed. While the authors ablated the extractor with GPT-OSS-120B, the grader, and evaluation judge remain entirely proprietary with no open-source alternatives tested. A second observation is that the win rates show much more improvement than the length-controlled. Not too much atention was drawn to this in the review process.

**Final Recommendation: Accept**

Justification: The paper makes a clear and practical contribution to rubric-based RL for LLM post-training, with a well-motivated method, and consistent gains across model scales and families. The reliance on proprietary models throughout the pipeline is a concern for reproducibility and broader applicability. Extending this to open source models for grading and evaluation is a natural next step. The gap between raw win rate and length-controlled (LC) improvements deserves more attention in the final manuscript, though other evaluation metrics like LC win rate still provide enough signal to recommend for acceptance.